# GIPO: Gaussian Importance Sampling Policy Optimization

**Chengxuan Lu** [1 2]   **Zhenquan Zhang** [1 2]   **Shukuan Wang** [1 2]   **Qunzhi Lin** [1 2]   **Baigui Sun** [† 1 2]   **Yang Liu** [† 1 2]

## Abstract

Post-training with reinforcement learning (RL) has recently shown strong promise for advancing multimodal agents beyond supervised imitation. However, RL remains limited by poor data efficiency, particularly in settings where interaction data are scarce and quickly become outdated. To address this challenge, GIPO (Gaussian Importance sampling Policy Optimization) is proposed as a policy optimization objective based on truncated importance sampling, replacing hard clipping with a log-ratio-based Gaussian trust weight to softly damp extreme importance ratios while maintaining non-zero gradients. Theoretical analysis shows that GIPO introduces an implicit, tunable constraint on the update magnitude, while concentration bounds guarantee robustness and stability under finite-sample estimation. Experimental results show that GIPO achieves state-of-the-art performance among clipping-based baselines across a wide range of replay buffer sizes, from near on-policy to highly stale data, while exhibiting superior bias–variance trade-off, high training stability and improved sample efficiency. Code is available at https://github.com/distanceLu/GIPO.

## 1. Introduction

In many real-world reinforcement learning (RL) applications, such as robotic control, healthcare decision-making, and industrial automation, interacting with the environment is prohibitively expensive or time-consuming. Consequently, strict on-policy learning, which requires fresh data for every parameter update, is often impractical due to limited throughput and data scarcity. To improve sample efficiency, training pipelines inevitably adopt asynchronous execution or heavy reliance on experience replay, reusing historical trajectories to maximize the utility of every interaction. However, this necessity introduces a fundamental challenge: the experience stored in the replay buffer is generated by older behavior policies that lag behind the current learner policy (Espeholt et al., 2018; Schulman et al., 2017). This *policy lag* creates a distribution mismatch, causing importance ratios between the target and behavior policies to exhibit heavy-tailed distributions as variability accumulates over time.

Two broad approaches are commonly used to handle this regime. One line adopts fully off-policy control (e.g. DQN/SAC-style actor–critic methods), which is explicitly designed to learn from replayed experience (Mnih et al., 2015; Haarnoja et al., 2018). Another line retains PPO-family conservative policy optimization as the backbone, extending it to replay-heavy settings to trade strict on-policy updates for higher throughput and practical stability (Espeholt et al., 2018; Meng et al., 2023). Given our target setting's large stochastic policies, sparse rewards, and high interaction costs, we follow the second route. Fully off-policy methods can be sensitive to critic learning instability under distribution shift, often requiring extensive tuning (Fujimoto et al., 2019; Kumar et al., 2020). By contrast, PPO-family objectives provide a mature, conservative surrogate with robust stability monitors (e.g., trust regions) and a direct actor update, making them a preferred backbone for reliable post-training.

However, standard stabilizers for policy lag do not necessarily translate into the effective reuse of stale replay. While target-side corrections (like V-trace(Espeholt et al., 2018)) improve value estimation, they do not address a distinct actor-side bottleneck. When importance ratios become heavy-tailed due to lag, standard PPO-style surrogates employ a "hard clipping" mechanism to constrain updates. In replay-heavy training, this mechanism often becomes overly aggressive, effectively zeroing out the gradient contribution of many stale samples. This leads to a phenomenon of utilization collapse: valuable but stale trajectories are replayed and processed computationally, yet they contribute almost nothing to the policy update, resulting in significant data inefficiency in cost-sensitive domains.

We therefore propose **GIPO** (**G**aussian **I**mportance sam-

[1]IROOTECH TECHNOLOGY, Hangzhou, China [2]Wolf 1069 b Lab, Sany Group, Hangzhou, China. Correspondence to: Baigui Sun <baigui.sun@irootech.com>, Yang Liu <yang.liu1@irootech.com>.

*Proceedings of the 43$^{rd}$ International Conference on Machine Learning*, Seoul, South Korea. PMLR 306, 2026. Copyright 2026 by the author(s).

pling **P**olicy **O**ptimization) to use stale replay more effectively. GIPO replaces the hard thresholding of standard methods with a smooth trust weighting in log-ratio space. Specifically, it applies a Gaussian kernel to the log-importance ratios, providing a symmetric, differentiable damping of extreme samples. This approach yields controlled down-weighting of off-policy data while preserving non-zero gradients in the tails. By doing so, GIPO allows stale replay to contribute small but informative updates rather than being effectively discarded, turning historical data into a useful signal without sacrificing stability.

Our contributions are as follows:

- We propose GIPO, a smooth log-ratio trust-weighted surrogate for PPO-style policy optimization designed to mitigate utilization collapse under policy lag. Furthermore, we demonstrate the extensibility of this framework through Advantage-Aware configurations that elegantly accommodate different trust region scales without sacrificing core log-space symmetry.

- Our theoretical analysis demonstrates that GIPO implicitly enforces a tunable bound on the policy update magnitude. We theoretically prove and empirically validate that GIPO achieves a superior bias-variance trade-off compared to baseline methods, providing formal guarantees of robustness and stability under finite-sample estimation.

- We evaluate GIPO across diverse complexities, from multi-seed evaluations on standard continuous control benchmarks to complex Meta-World tasks and the LIBERO benchmark (utilizing a 7B OpenVLA-OFT backbone). This large-scale study (consuming over 10,000 H200 GPU-hours) demonstrates that GIPO achieves superior sample efficiency and effective replay usage compared to clipping-based baselines, particularly when data freshness is limited.

## 2. Related Work

In scalable policy learning, policy lag occurs when experience is generated by a stale behavior policy $\mu$ relative to the learner $\pi_\theta$, widespread in distributed systems or when reusing rollouts (Schulman et al., 2017; Espeholt et al., 2018). Under such effectively off-policy updates, heavy-tailed importance ratios often cause optimization instability and hinder the utilization of stale replay. Existing work primarily employs two approaches to address this: (i) target-side off-policy correction via truncated importance sampling, and (ii) surrogate-side ratio control. A broader related line studies trajectory reuse via explicit variance bounding (Metelli et al., 2018; 2020; Tirinzoni et al., 2019; Lin et al., 2025).

### 2.1. Truncated Off-policy Correction under Policy Lag

A classical response to behavior-target mismatch is importance sampling (IS), but naive IS can incur prohibitive variance when $\mu$ and $\pi_\theta$ diverge. A widely adopted principle is therefore truncated importance sampling, which trades controlled bias for substantially reduced variance. Retrace($\lambda$) is a canonical example for multi-step bootstrapping under off-policy drift: it uses per-decision truncated ratios in the return recursion, with a $\lambda$-style trace that geometrically down-weights distant steps, yielding a stable bias–variance trade-off for multi-step value targets (Munos et al., 2016). In distributed actor-learner training, IMPALA introduces V-trace, which applies truncated per-decision ratios when constructing multi-step value targets and the corresponding policy-gradient advantages, explicitly motivated by actor-learner delay (Espeholt et al., 2018). Concretely, V-trace uses two clipped coefficients: one for policy-gradient correction and one for trace propagation, so that both the immediate correction and the multi-step credit assignment remain bounded. Replay-heavy actor-critic methods such as ACER further integrate truncation with bias correction and conservative updates to safely reuse stale trajectories (Wang et al., 2017).

While these methods substantially improve the reliability of value and advantage estimation under policy lag, they primarily act on the target construction side. Under heavy-tailed ratios, conservative policy surrogates can still yield weak actor updates on stale replay. Even with stabilized advantages, tail-ratio samples may be strongly down-weighted or enter saturated regions under ratio-based objectives. Consequently, the effective update may become dominated by a small subset of transitions due to contribution degeneracy. This motivates complementary modifications at the surrogate level.

### 2.2. Beyond Hard Clipping: Smooth Surrogates and Off-policy PPO

PPO (Schulman et al., 2017) popularized conservative policy optimization via a clipped surrogate objective. Intuitively, clipping imposes a hard trust region on the importance ratio: when a ratio moves far beyond the clipping interval, the surrogate switches to a clipped branch so that further changes in the ratio no longer increase the objective. Consequently, for stale replay where tail ratios are frequent, many samples can fall into this saturated regime. Their effective influence on the policy update becomes strongly limited. This observation has motivated a line of work that replaces hard clipping with smoother, continuously varying ratio control.

For example, smooth or soft clipping methods such as SCAPPO (Wang et al., 2023) replace the discontinuous switch induced by hard thresholds with differentiable gating, so tail samples are down-weighted gradually rather than

abruptly. Similarly, SAPO (Gao et al., 2025) proposes a smooth advantage-weighted objective to handle importance ratios. While SAPO distinguishes between positive and negative advantages by applying different decay, GIPO can also implement this advantage-aware weighting by varying its damping scale $\sigma$. Crucially, even with different scales for positive and negative advantages, GIPO still maintains its core log-space symmetry. Other PPO variants aim to better approximate trust-region behavior by refining rollback rules or incorporating additional constraints, improving robustness when ratios drift (Wang et al., 2020; Cheng et al., 2022). Meanwhile, off-policy extensions of PPO explicitly combine replay with conservative ratio control, demonstrating that PPO-style objectives can be adapted to replay-heavy training (Meng et al., 2023).

A complementary line enforces conservatism by regularizing distribution shift directly, for example by penalizing divergences between the updated policy and a behavior policy, or by introducing trust-region constraints in distribution space (Wang et al., 2019; Touati et al., 2020). Such divergence-based formulations provide continuous control over update magnitude without relying solely on hard ratio thresholds. However, they are typically evaluated through aggregate stability measures such as KL divergence and do not directly quantify whether stale replay continues to contribute meaningful actor gradients under severe lag.

Overall, existing surrogate-side approaches largely focus on improving the robustness of updates under heavy ratio tails. While methods like SAPO introduce smooth weighting, they often lack a mechanism to explicitly balance the bias–variance trade-off in a symmetric and theoretically consistent manner. Our work targets this gap by combining smooth ratio damping with utilization diagnostics that directly measure stale-replay gradient contribution, ensuring effective learning from historical data while maintaining stability. In parallel, IS-based policy optimization studies trajectory reuse with estimator-side variance/bound control (e.g., high-confidence surrogates, per-decision IS, and multiple importance sampling) (Metelli et al., 2018; 2020; Tirinzoni et al., 2019), and recent analyses extend such reuse ideas to natural policy gradient via importance weighting (Lin et al., 2025). These lines clarify how to control IS variance under reuse, but they do not directly quantify the actor-update utilization of stale replay in replay-heavy actor–critic training.

## 3. Preliminaries

We consider the problem of policy optimization in off-policy, replay-heavy actor–critic settings. In this regime, the learner must effectively utilize historical trajectories collected by possibly stale behavior policies. This section formalizes the problem setup, defines the heavy-tailed importance ratios

caused by policy lag, and specifically outlines the limitations of standard clipped surrogates under these conditions.

### 3.1. MDP and Policy Optimization

We formulate the control problem as a discounted Markov Decision Process (MDP) defined by the tuple $\langle \mathcal{S}, \mathcal{A}, P, r, \gamma \rangle$, where $\gamma \in (0, 1)$ is the discount factor. A stochastic policy $\pi_\theta(a \mid s)$, parameterized by $\theta$, interacts with the environment to generate trajectories. The goal is to maximize the expected discounted return:

$$J(\pi_\theta) = \mathbb{E}_{\tau \sim \pi_\theta} \left[ \sum_{t=0}^{\infty} \gamma^t r(s_t, a_t) \right] \quad (1)$$

We adopt an actor–critic architecture where a critic $V_\phi(s)$ estimates the state-value function to reduce the variance of policy gradient updates via advantage estimation.

### 3.2. Replay-Heavy Training and Policy Lag

In distributed or replay-heavy learning, trajectory segments are not collected by the current policy $\pi_\theta$, but by a *behavior policy* $\mu$. These segments are stored in a replay buffer $\mathcal{B}$. Due to asynchronous data collection, delayed synchronization, or the reuse of data across multiple training epochs, there is an inevitable *policy lag* between the learner $\pi_\theta$ and the behavior $\mu$ (i.e., $\mu \neq \pi_\theta$).

We quantify this discrepancy using the per-step importance ratio:

$$\rho_t \triangleq \frac{\pi_\theta(a_t \mid s_t)}{\mu(a_t \mid s_t)}. \quad (2)$$

In replay-heavy scenarios, $\rho_t$ frequently drifts far from 1, exhibiting a heavy-tailed distribution that poses significant challenges for stability.

### 3.3. Standard PPO and Utilization Collapse

To maintain stability, the Proximal Policy Optimization (PPO) algorithm(Schulman et al., 2017) restricts the update size using a clipped surrogate objective:

$$\mathcal{L}^{\text{CLIP}}(\theta) = -\mathbb{E}_{(s_t, a_t) \sim \mathcal{B}} \Big[ \min \big( \rho_t A_t, \\ \text{clip}(\rho_t, 1 - \epsilon, 1 + \epsilon) A_t \big) \Big], \quad (3)$$

where $A_t$ is the estimated advantage and $\epsilon$ is a hyperparameter.

While effective for near-on-policy data, this hard clipping mechanism creates a utilization bottleneck under heavy policy lag. When stale replay causes $\rho_t$ to fall outside the clipping interval $[1 - \epsilon, 1 + \epsilon]$, the gradient contribution of that sample becomes zero (if the advantage sign dictates a move further outside the interval). We refer to this phenomenon

as utilization collapse, where a large fraction of the replay buffer is theoretically valid but numerically ignored by the optimizer.

## 3.4. Surrogate-side vs. Target-side Correction

Handling off-policy data involves two components: target-side correction (estimating accurate values and advantages $A_t$, e.g., via V-trace) and surrogate-side weighting (weighting the policy gradient $\nabla_\theta \log \pi_\theta$ based on $\rho_t$). In this work, we assume standard target-side corrections (e.g., Generalized Advantage Estimation (Schulman et al., 2016)) are in place. We focus on the surrogate-side, proposing a mechanism to replace the hard clipping in Eq. (3) to recover gradient utilization from stale data.

# 4. Gaussian Importance Sampling Policy Optimization

To address the utilization collapse observed in replay-heavy PPO, we propose GIPO, which replaces PPO's hard piecewise clipping with a smooth, continuously differentiable trust mechanism in log-ratio space. This approach allows the learner to softly down-weight stale samples with extreme importance ratios while preserving non-zero gradients for valid updates.

## 4.1. Log-Space Gaussian Weighting

The core of our method is a damping weight that evaluates the freshness or reliability of a sample based on its importance ratio. Let $\rho_t(\theta)$ be the importance ratio as defined in Eq. (2). We first define a detached ratio to prevent gradients from propagating through the weighting mechanism:

$$\bar{\rho}_t \triangleq \text{sg}(\rho_t(\theta)) \tag{4}$$

where $\text{sg}(\cdot)$ denotes the stop-gradient operator.

We then define the Gaussian trust weight $\omega(\bar{\rho}_t; \sigma)$ based on the distance of $\log(\bar{\rho}_t)$ from 0 (i.e., $\rho_t = 1$):

$$\omega(\bar{\rho}_t; \sigma) \triangleq \exp\left(-\frac{1}{2}\left(\frac{\log(\bar{\rho}_t)}{\sigma}\right)^2\right). \tag{5}$$

Here, $\sigma > 0$ is a unified scale parameter controlling the strength of the damping. A smaller $\sigma$ enforces a stricter trust region (concentrating weight around $\rho \approx 1$), while a larger $\sigma$ tolerates larger deviations. For numerical stability in implementation, we clamp $\bar{\rho}_t$ within fixed bounds $[\rho_{\min}, \rho_{\max}]$ before computing the logarithm, ensuring the weight remains well-defined.

## 4.2. The GIPO Surrogate Objective

We incorporate the Gaussian weight into the policy optimization objective. Unlike PPO, which clips the surrogate,

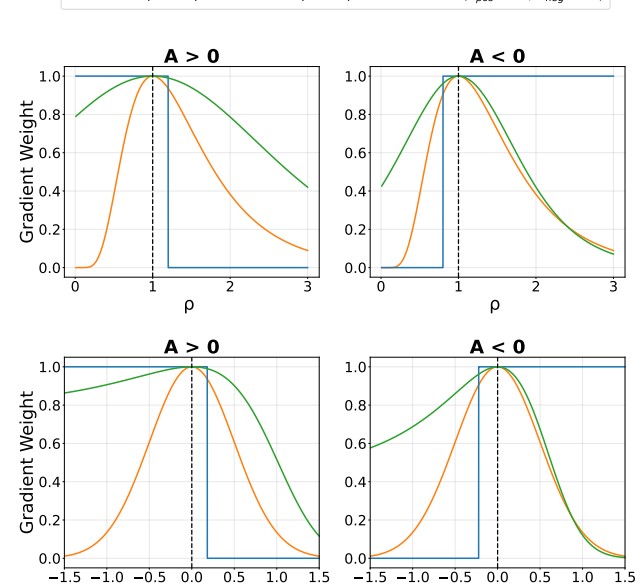

*Figure 1.* **Comparison of gradient weights.** Top: Weights vs. importance ratio $\rho$. Bottom: Weights vs. $\log(\rho)$. The log-scale plots highlight GIPO's unique symmetry ($\omega(\rho) = \omega(1/\rho)$) compared to PPO and SAPO, ensuring balanced updates for equivalent deviations.

GIPO minimizes a weighted importance sampling loss:

$$\mathcal{L}_\pi^{\text{GIPO}}(\theta) = -\mathbb{E}_{(s_t, a_t) \sim \mathcal{B}}\left[\omega(\bar{\rho}_t; \sigma) \cdot \rho_t(\theta) \cdot A_t\right]. \tag{6}$$

Since $\omega$ depends only on $\bar{\rho}_t$, it is treated as a constant scalar coefficient during backpropagation. The gradient of this objective with respect to $\theta$ is:

$$\nabla_\theta \mathcal{L}_\pi^{\text{GIPO}}(\theta) = -\mathbb{E}_{(s_t, a_t) \sim \mathcal{B}}$$
$$\left[\underbrace{\omega(\bar{\rho}_t; \sigma)\rho_t}_{m_t} \nabla_\theta \log \pi_\theta(a_t \mid s_t) A_t\right]. \tag{7}$$

The term $m_t \triangleq \omega(\bar{\rho}_t; \sigma)\rho_t$ acts as the effective surrogate multiplier. It determines the magnitude of the update for each sample. This formulation ensures that the policy gradient is scaled continuously based on how far the behavior policy deviates from the current policy.

## 4.3. Desirable Properties

The design of the Gaussian weight $\omega$ introduces several desirable properties for off-policy training.

**Symmetric Trust and Boundedness.** As visualized in Figure 1, standard PPO clipping is asymmetric; it penalizes ratios differently depending on the arithmetic distance from 1. In contrast, GIPO is strictly symmetric in log-space:

$$\omega(\rho; \sigma) = \omega(1/\rho; \sigma). \tag{8}$$

This implies that a sample where the target policy is $k$ times more likely than the behavior ($\rho = k$) is treated with the same reliability weight as a sample where it is $k$ times less likely ($\rho = 1/k$). Crucially, this symmetry enforces bounded effective updates at both asymptotic limits. As $\rho_t \to \infty$, the Gaussian decay dominates the linear growth of the ratio to prevent gradient explosion. Conversely, as $\rho_t \to 0$, the weight vanishes to accelerate the suppression of irrelevant samples. Consequently, GIPO naturally filters out stale data exhibiting extreme distribution shifts in either direction, treating them as unreliable noise without requiring hard thresholds.

**Smoothness.** Unlike the piecewise-defined PPO surrogate, which has non-differentiable points at the clipping boundaries, GIPO's weight $\omega$ is smooth for all $\rho > 0$. This removes the discontinuous switching behavior where a small perturbation in $\rho_t$ could abruptly toggle a sample's contribution between zero and non-zero. Smoothness improves optimization stability and facilitates more effective utilization of samples that lie just outside the trust region, allowing them to contribute with dampened but non-zero gradients rather than being discarded entirely.

**Bias–Variance Interpolation.** The scale $\sigma$ interpolates smoothly between on-policy and off-policy regimes, governing an explicit bias–variance trade-off. As $\sigma \to 0$, the Gaussian weight collapses to a spike at $\rho = 1$, so only samples with $\rho_t \approx 1$ contribute; the estimator degenerates toward using only near-on-policy samples, yielding a highly conservative but biased update, as it effectively ignores the distribution shift inherent in off-policy data. Conversely, as $\sigma \to \infty$, $\omega(\bar{\rho}_t; \sigma) \to 1$, and the objective asymptotically recovers the standard importance-sampling policy gradient, yielding an unbiased estimator for off-policy data but suffering from high (often unbounded) variance.

### 4.4. Overall Algorithm

We optimize the actor and critic jointly. The total objective function is:

$$\min_{\theta, \phi} \mathcal{L}(\theta, \phi) = \mathcal{L}_\pi^{\text{GIPO}}(\theta) + \lambda_v \mathcal{L}_v(\phi) - \lambda_h \mathcal{L}_{\text{ent}}(\theta), \quad (9)$$

$$\mathcal{L}_v(\phi) = \mathbb{E}_{s_t \sim \mathcal{B}}\left[\left(V_\phi(s_t) - v_t^{\text{trace}}\right)^2\right], \quad (10)$$

$$\mathcal{L}_{\text{ent}}(\theta) = \mathbb{E}_{s_t \sim \mathcal{B}}\left[\mathcal{H}\left(\pi_\theta(\cdot \mid s_t)\right)\right], \quad (11)$$

where $\mathcal{L}_v$ is the squared error loss for the value function (e.g., against GAE or TD($\lambda$) targets) and $\mathcal{L}_{\text{ent}}$ is an entropy bonus to encourage exploration. The complete training procedure is summarized in Algorithm 1 in Appendix A.

## 5. Theory Foundation

In this section, we adopt the standard RL setup and the objective $J(\pi)$ (omitting $\theta$ for notation clarity) defined in (1). Following the monotonic improvement principle of TRPO (Schulman et al., 2015; Kakade & Langford, 2002), which ensures $J(\pi') \geq J(\pi)$ by maximizing a surrogate, we introduce the normalized discounted state occupancy distribution $d_\pi(s)$ defined as:

$$d_\pi(s) := (1 - \gamma) \sum_{t=0}^{\infty} \gamma^t \Pr_\pi(s_t = s). \quad (12)$$

Based on (12), $J(\pi)$ can be rewritten as:

$$J(\pi) = \frac{1}{1 - \gamma} \mathbb{E}_{s \sim d_\pi, a \sim \pi}[r(s, a)]. \quad (13)$$

To address the off-policy regime (Meng et al., 2022), GIPO replaces the raw importance weight $\rho'_t = \frac{\pi'(a|s)}{\mu(a|s)}$ with an effective weight $\omega(\bar{\rho}'_t; \sigma) \, \rho'_t$. Omitting the subscript $t$, we obtain the attenuated surrogate evaluated under $d_\mu$:

$$\tilde{L}(\pi') \triangleq J(\pi) + \frac{1}{1 - \gamma} \mathbb{E}_{s \sim d_\mu, \, a \sim \mu(\cdot|s)} \\ \left[\omega(\bar{\rho}'; \sigma) \, \rho' \, A_\pi(s, a)\right]. \quad (14)$$

The complete derivations are given in Appendix B.1.

### 5.1. Monotonic Improvement Guarantee for GIPO

The surrogate (27) is generally *biased* because $\omega(\bar{\rho}'; \sigma) \, \rho'$ differs from the canonical importance weight $\rho'$. One can further decompose the off-policy surrogate into the proposed attenuated surrogate and a separate bias term:

$$\mathbb{E}_{d_\mu, \mu}[\rho' A_\pi] = \mathbb{E}_{d_\mu, \mu}[\omega(\bar{\rho}'; \sigma) \, \rho' \, A_\pi] + \\ \underbrace{\mathbb{E}_{d_\mu, \mu}[(1 - \omega(\bar{\rho}'; \sigma)) \, \rho' \, A_\pi]}_{\text{bias term}}. \quad (15)$$

Using $\mathbb{E}[|X|] \geq |\mathbb{E}[X]|$ and the bounded advantage assumption (25), the bias term admits:

$$\left| \mathbb{E}_{d_\mu, \mu}[(1 - \omega(\bar{\rho}'; \sigma)) \, \rho' \, A_\pi] \right| \\ \leq \varepsilon \, \mathbb{E}_{s \sim d_\mu, \, a \sim \mu(\cdot|s)}[(1 - \omega(\bar{\rho}'; \sigma)) \, \rho'] \\ = \varepsilon \, \mathbb{E}_{s \sim d_\mu, \, a \sim \pi'(\cdot|s)}[(1 - \omega(\bar{\rho}'; \sigma))]. \quad (16)$$

**Lemma 5.1.** *For any $\tau > 0$,*

$$\mathbb{E}_{s \sim d_\mu, \, a \sim \pi'(\cdot|s)}[(1 - \omega(\bar{\rho}'; \sigma))] \\ \leq \frac{\tau^2}{2\sigma^2} + 2e^{-\tau} + \sqrt{\frac{\delta}{2}}. \quad (17)$$

Where $\delta$ denotes $D_{\mathrm{KL}}^{\max}(\mu, \pi')$. Lemma 5.1 provides an upper bound using only per-state trust region. Full step-by-step derivations are in Appendix B.2.

Using $\mathbb{E}[X] \geq -|\mathbb{E}[X]|$, Lemma 5.1 together with (15) yields the following lower bound for the expected performance $J(\pi')$.

**Theorem 5.2** (Lower bound for the expected performance). *Assume advantage is bounded as described in (25) and define $\delta$ as $D_{\mathrm{KL}}^{\max}(\mu, \pi')$. For any $\tau > 0$,*

$$J(\pi') \geq \tilde{L}(\pi') - C\Big( D_{\mathrm{KL}}^{\max,sqrt}(\mu, \pi)\, D_{\mathrm{KL}}^{\max,sqrt}(\pi, \pi')$$
$$+ D_{\mathrm{KL}}^{\max}(\pi, \pi')\Big) - \frac{\varepsilon}{1-\gamma}\left(\frac{\tau^2}{2\sigma^2} + 2e^{-\tau} + \sqrt{\frac{\delta}{2}}\right),$$
(18)

*where $C = \frac{4\gamma\varepsilon}{(1-\gamma)^2}$ .*

In practice, we do not explicitly optimize the analytic lower bound, as its penalty terms depend solely on the trust-region radius (and other fixed constants). Instead, maximizing the surrogate objective $\tilde{L}$ monotonically improves the lower-bound certificate (RHS of (18)) up to a bounded slack, while implicitly enforcing a soft trust-region constraint. This approach can be regarded as equivalently optimizing the lower-bound certificate to a certain extent, thereby forming a more tractable objective.

### 5.2. Finite-Sample Control of the Surrogate

A key challenge in off-policy policy optimization is reliably estimating the improvement surrogate from finite samples. Standard importance sampling suffers from unbounded variance, undermining finite-sample reliability. While PPO mitigates this via hard clipping, it introduces non-differentiability and discards useful information beyond the clipping threshold. GIPO instead employs Gaussian-like attenuation to induce bounded effective weights, enabling the use of concentration inequalities such as Hoeffding's. This yields high-probability confidence bounds on the surrogate estimate, forming a principled basis for certified policy improvement under limited data, with improved stability and theoretical guarantees.

**Lemma 5.3** (Global bound on $\omega(\bar{\rho}'; \sigma)\rho'$). *For all $\rho' > 0$,*

$$0 < \omega(\bar{\rho}'; \sigma)\rho' \leq \exp\left(\frac{\sigma^2}{2}\right). \tag{19}$$

*Moreover, the upper bound is achieved at $\rho' = \exp(\sigma^2)$.*

**Theorem 5.4.** *Assume samples $(s_i, a_i) \sim d_\mu, \mu(\cdot \mid s)$ and define*

$$\widehat{\tilde{L}}(\pi') := \frac{1}{N}\sum_{i=1}^{N} \omega(\bar{\rho}'; \sigma)\rho'\, A_\pi(s_i, a_i). \tag{20}$$

*By Lemma 5.3 and* (25)*, each summand is bounded by $\varepsilon e^{\sigma^2/2}$ in absolute value.*

*Under the above assumptions, for any $\alpha \in (0, 1)$, with probability at least $1 - \alpha$,*

$$\left|\widehat{\tilde{L}}(\pi') - \tilde{L}(\pi')\right| \leq \varepsilon e^{\sigma^2/2} \sqrt{\frac{2\ln(2/\alpha)}{N}}. \tag{21}$$

(21) quantifies the *statistical noise* incurred when $\tilde{L}(\pi')$ is estimated from a finite batch. In particular, the deviation term on the right-hand side of (21) is an explicit high-probability upper bound on the error due to sampling randomness. Consequently, any update rule that uses $\widehat{\tilde{L}}(\pi')$ must account for this noise: a candidate policy $\pi'$ that appears beneficial under the empirical estimate may fail to improve in expectation if the empirical gain is smaller than the deviation bound.

## 6. Experiments and Results

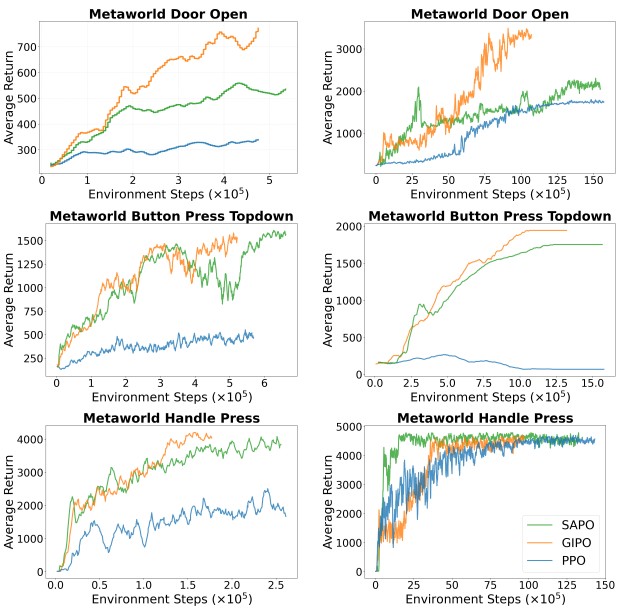

*Figure 2.* **Learning curves on the Meta-World benchmark under Stale and Fresh regimes.** Left: Stale; Right: Fresh. Curves show episodic average return over environment steps from a single run per configuration.

In this section, we evaluate GIPO across varying levels of complexity. We first analyze its learning behavior and resilience to policy lag on the Meta-World benchmark. Next, we scale our evaluation to the LIBERO benchmark utilizing a 7B-parameter OpenVLA-OFT backbone to demonstrate its effectiveness in large-scale instruction-conditioned manipulation. Finally, we provide a quantitative bias–variance analysis using an analytically tractable GridWorld environment to dissect its theoretical advantages. Due to space

constraints, extensive multi-seed evaluations on MuJoCo and Classic Control, alongside targeted ablation studies on the Advantage-Aware GIPO framework, are deferred to Appendix D.

## 6.1. Experiment Settings

We study replay-heavy PPO-style policy optimization under an asynchronous actor–learner pipeline. Experience is collected by rollout actors, written to a replay buffer, and later consumed by trainers for learning. This decoupling induces policy lag and makes replay off-policy relative to the current learner. Implementation details and training configurations are reported in the Appendix C.2.

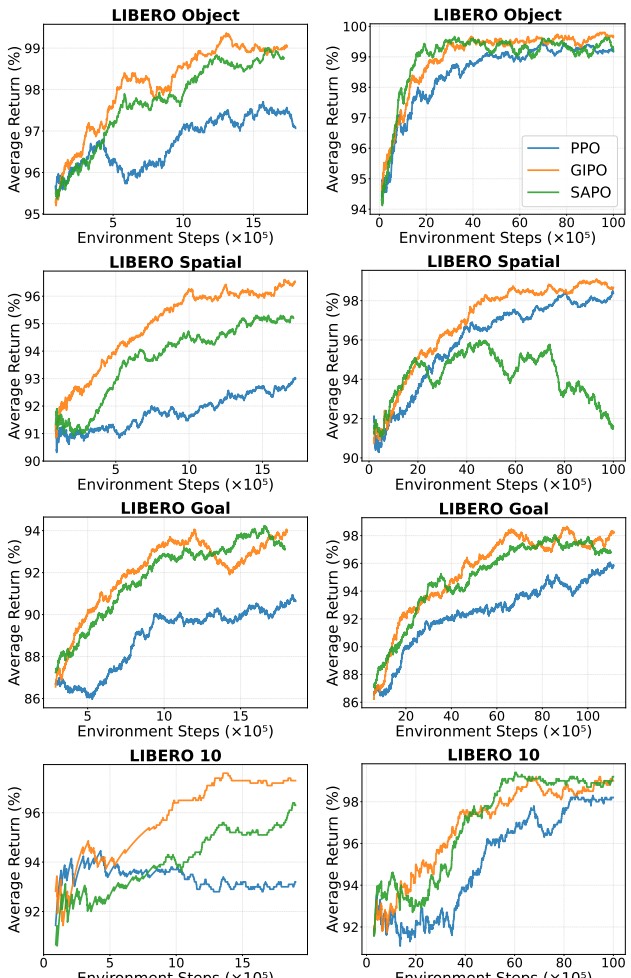

*Figure 3*. **Learning curves on LIBERO suites under Stale and Fresh regimes.** Rows from top to bottom: LIBERO-Object, LIBERO-Spatial, LIBERO-Goal, and LIBERO-10. Left: Stale; Right: Fresh. Curves show episodic average return over environment steps from a single run per configuration.

**Baselines** Unless stated otherwise, methods share the same backbone, replay configuration, and target and ad-

*Table 1*. Mean Normalized Scores (Aggregated via IQM over 10 Meta-World tasks with 5 random seeds) in the Stale regime. Brackets denote the damping scales ($\sigma_{\text{pos}}, \sigma_{\text{neg}}$) for positive and negative advantages, respectively.

| RANK | ALGORITHM | IQM SCORE |
|------|-----------|-----------|
| 1 | **GIPO (1.0, 1.0)** | **0.730** |
| 2 | GIPO (0.5, 0.5) | 0.589 |
| 3 | GIPO (1.0, 0.5) | 0.543 |
| 4 | GIPO (0.2, 0.2) | 0.542 |
| 5 | GIPO (0.5, 0.25) | 0.477 |
| 6 | GIPO (0.2, 0.1) | 0.445 |
| 7 | SAPO | 0.412 |
| 8 | PPO | 0.180 |

vantage construction. We compare actor-side ratio handling strategies that differ only in the surrogate weight applied to the policy-gradient term: PPO-Clip(Schulman et al., 2017), SAPO(Gao et al., 2025), and GIPO with multiple damping scales.

**Fresh vs. stale regimes** We construct two regimes by changing rollout throughput while keeping the replay configuration and sampling strategy fixed. The fresh regime collects experience at higher throughput than the stale regime, so replay stays closer to the current learner. The stale regime reduces throughput, which increases the fraction of replay generated by older behavior policies.

To quantify lag, each replay entry stores a behavior-policy version identifier. Let $\Delta v_t$ denote the version gap between the current learner policy version and the behavior policy associated with sample $t$. We summarize staleness using OldFrac and OldGapP95, and summarize tail behavior using the drift statistic $D_{0.95}$ defined in Eq. (40). Appendix C.3 reports the full regime specification, the threshold used for the old-sample split, and aggregated statistics with representative training-time trajectories.

### 6.2. Meta-World: Main learning outcomes across tasks

We report episodic average return as the primary learning outcome. Figure 2 shows learning curves over environment steps for three Meta-World (Yu et al., 2020) tasks, with stale regime in the left column and fresh regime in the right column.

In these traces, separation between ratio-handling strategies is larger in the stale regime, where replay contains many older samples. In the reported runs, GIPO attains higher episodic returns than PPO-Clip on the displayed tasks and is also higher than SAPO in these traces. The differences are reflected in earlier progress and a higher return level toward the end of training. In contrast, PPO-Clip often saturates earlier at a lower return level in the same stale setting. In

the fresh regime, methods are closer on some tasks and the ordering is more task-dependent.

To interpret these gaps under stale replay, Appendix C.5 reports utilization diagnostics on a representative task. These diagnostics indicate that hard clipping yields a higher near-zero contribution fraction, while smooth ratio handling is associated with higher effective utilization of old replay in the same regime, consistent with the learning-curve patterns in Figure 2.

To ensure statistical validity beyond individual tasks, we expand our evaluation to an aggregated setting across 10 Meta-World tasks using 5 random seeds per configuration (totaling 400 training runs in the Stale regime). As reported in Table 1, we evaluate GIPO across multiple Unified Damping Scales ($\sigma$), as well as its Advantage-Aware variants that apply different scales for positive and negative advantages (e.g., $\sigma_{\text{neg}} = 0.5 \cdot \sigma_{\text{pos}}$).

The aggregated Interquartile Mean (IQM) scores show that all GIPO variants consistently outperform the PPO and SAPO baselines under policy lag. Specifically, GIPO (1.0, 1.0) achieves the highest score of 0.730, outperforming both SAPO and PPO by a large margin. Furthermore, the competitiveness of the Advantage-Aware configurations (such as GIPO (1.0, 0.5)) demonstrates that our framework is highly robust to parameter choices and can flexibly incorporate advantage-conditioned weighting while maintaining performance.

### 6.3. Experiments on LIBERO Benchmark

To evaluate the effectiveness of GIPO in multi-task robotic manipulation, we conduct extensive experiments on the LIBERO (Liu et al., 2023) dataset using the OpenVLA-OFT (Kim et al., 2025) backbone. The total computational effort for our LIBERO evaluation alone exceeds 10,000 H200 GPU-hours, involving the processing of over 730 million interactive samples through a 7B-parameter VLA backbone. The consistency of GIPO across this massive computational scale, spanning 8 distinct scenarios and 24 full training curves, provides a level of empirical validation that is rarely feasible when training billion-parameter VLA models in computationally demanding environments.

As illustrated in Figure 3, GIPO consistently outperforms both PPO and SAPO in convergence speed and sample efficiency across all suites. On LIBERO-Spatial and LIBERO-Goal, GIPO reaches a near-optimal success rate much earlier, at around 1M environment steps, than the baselines. On the more complex LIBERO-10 suite, GIPO also shows a steady upward trend and finishes with a higher average return, while the baselines improve more slowly and fluctuate more.

### 6.4. Analytic Bias–Variance Study

Furthermore, we investigate the robustness of our algorithm regarding data freshness, which refers to the magnitude of the off-policy shift during training. We conducted two sets of experiments by varying the ratio of rollout actors to trainers: a high-freshness configuration (10 actors per trainer) and a low-freshness configuration (3 actors per trainer). In the 3-actor scenario, the lower data collection rate relative to the optimization frequency results in the trainer utilizing "staler" data from the replay buffer. As shown in Figure 3, the performance gap between algorithms becomes significantly more pronounced in this low-freshness regime. While the performance differences are relatively minor when data is fresh (10 actors), GIPO demonstrates superior robustness as data staleness increases, maintaining high performance and stability where baselines show significant degradation. This indicates that GIPO's optimization objective is more effective at managing the distribution shift inherent in off-policy data, making it highly suitable for practical robotic training scenarios where high-throughput data collection is computationally or physically restricted.

To quantitatively analyze the bias and variance of different clipping methods under varying degrees of policy lag, we utilize an analytically tractable toy environment: a $2 \times 2$ GridWorld, where $S_0$ is the initial state and $S_G$ is the absorbing goal state (Figure 4). Additionally, we construct a target policy $\pi$ and three behavior policies $\mu$ to simulate different levels of data staleness. Case A represents a high-lag scenario where the behavior policy $\mu$ is v. As we transition to Case B and Case C, $\mu$ assigns progressively more probability to $\pi$, thereby policy lag decreases. Details of experimental settings are provided in Appendix C.8.

Because this MDP is fully enumerable, we can compute the exact expectation and variance of the gradient estimator $g$ via action enumeration based on Bellman equations, without Monte Carlo sampling noise. The bias is defined as the deviation of $\mathbb{E}_\mu[g]$ from the ground-truth gradient, and variance is defined as $\text{Var}_\mu[g]$.

Figure 4 illustrates the bias–variance distributions and the corresponding Pareto frontiers for the evaluated methods. Overall, as the behavior policy $\mu$ transitions from Case A to Case C, i.e., as policy lag decreases, the bias gradually decreases, whereas the overall variance level exhibits an increasing trend.

GIPO achieves a superior bias–variance trade-off compared to No-Clip, PPO, and SAPO, establishing a dominant Pareto frontier through the tunable parameter $\sigma$, where smaller values suppress variance from heavy-tailed ratios while larger values yield nearly unbiased updates. This flexibility allows GIPO to consistently approximate the optimal frontier across varying degrees of policy lag. In contrast, SAPO

proves less robust and is dominated in Cases A and B, whereas PPO's observed zero variance in these regimes is merely an artifact of full gradient clipping. Indeed, in Case C where valid signals exist, PPO falls short of GIPO. These results align well with our analytical derivations, demonstrating that GIPO utilizes stale data more stably and efficiently in scenarios involving policy lag.

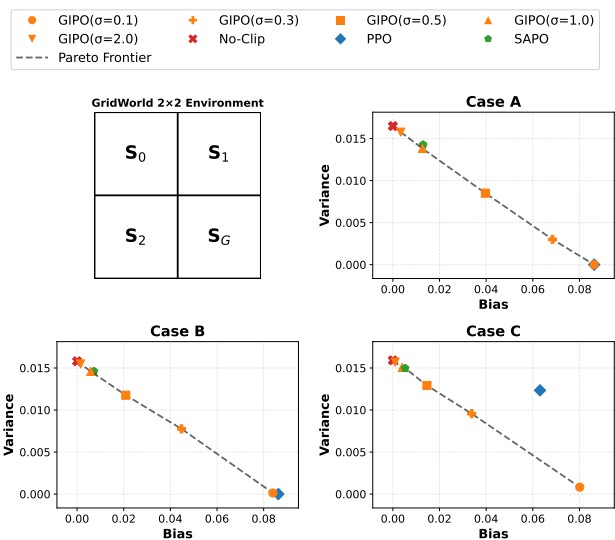

*Figure 4.* **Bias–variance trade-off in 2×2 GridWorld.** Case A represents high policy lag, Case B represents moderate policy lag, and Case C represents low policy lag. The dashed line indicates the Pareto frontier derived by GIPO.

## 7. Conclusion and Future Work

Focusing on replay-heavy training, we identified that PPO's hard clipping inefficiently discards stale data with heavy-tailed importance ratios. To resolve this, we introduced GIPO, which replaces clipping with a smooth Gaussian trust weight in log-ratio space to continuously damp extreme ratios. Our results demonstrate that standard GIPO significantly improves the utilization of stale replay while maintaining robust stability. Furthermore, we demonstrated the extensibility of our framework through an Advantage-Aware configuration, which applies different trust region scales based on the advantage sign to flexibly extract further performance gains without sacrificing core log-space symmetry. Future directions will focus on validating GIPO's real-world efficiency in physical robot post-training scenarios with inevitable replay staleness.

## Impact Statement

This paper presents work whose goal is to advance the field of Machine Learning. There are many potential societal consequences of our work, none which we feel must be specifically highlighted here.

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

# A. Algorithm Pseudocode

---

**Algorithm 1** GIPO learner update (one iteration)

---

**Require:** Replay buffer $\mathcal{B}$; policy $\pi_\theta$; value function $V_\phi$; discount $\gamma$; GIPO scale $\sigma$; clipping bounds $(\rho_{\min}, \rho_{\max})$ for $\log(\cdot)$; loss weights $(\lambda_v, \lambda_h)$.

1: Sample a batch $\{(s_t, a_t, r_t, s_{t+1}, \mu(a_t \mid s_t))\}$ from $\mathcal{B}$.
2: Compute ratios $\rho_t \leftarrow \pi_\theta(a_t \mid s_t)/\mu(a_t \mid s_t)$.
3: Obtain an advantage estimate $\hat{A}_t$ for each transition.
4: Form stop-gradient ratios $\bar{\rho}_t \leftarrow \mathrm{sg}(\rho_t)$.
5: Compute trust weights $\omega_t \leftarrow \omega(\mathrm{clip}(\bar{\rho}_t, \rho_{\min}, \rho_{\max}); \sigma)$.
6: Compute multiplier $m_t \leftarrow \omega_t \rho_t$ and proxy $u_t \leftarrow |m_t \hat{A}_t|$ for logging.
7: Compute policy loss $\mathcal{L}_\pi \leftarrow -\frac{1}{|\mathcal{D}|} \sum_t m_t \hat{A}_t$.
8: Compute value loss $\mathcal{L}_v \leftarrow \frac{1}{|\mathcal{D}|} \sum_t \left(V_\phi(s_t) - \hat{v}_t\right)^2$ (target $\hat{v}_t$ per implementation).
9: Compute entropy bonus $\mathcal{L}_{\mathrm{ent}} \leftarrow \frac{1}{|\mathcal{D}|} \sum_t \mathcal{H}(\pi_\theta(\cdot \mid s_t))$.
10: Update parameters $(\theta, \phi)$ by minimizing $\mathcal{L} \leftarrow \mathcal{L}_\pi + \lambda_v \mathcal{L}_v - \lambda_h \mathcal{L}_{\mathrm{ent}}$.

---

# B. Additional Proofs

## B.1. Derivation of GIPO's Surrogate

In Trust Region Policy Optimization (TRPO)(Schulman et al., 2015), a standard first-order surrogate replaces the true performance difference by the advantage averaged under the reference occupancy $d_\pi$ and the candidate policy $\pi'$:

$$L^{\text{TRPO}}(\pi') = \mathbb{E}_{s\sim d_\pi,\, a\sim\pi'(\cdot|s)}\big[A_\pi(s,a)\big]. \tag{22}$$

Using importance sampling to express the $\pi'$-expectation with samples from $\pi$ yields

$$L^{\text{TRPO}}(\pi') = \mathbb{E}_{s\sim d_\pi,\, a\sim\pi(\cdot|s)}\Big[\frac{\pi'(a\mid s)}{\pi(a\mid s)}\, A_\pi(s,a)\Big]. \tag{23}$$

where $\frac{\pi'(a|s)}{\pi(a|s)}$ is the importance sampling ratio.

TRPO introduces a surrogate objective function to approximate the expected return $J(\pi')$ of a new policy $\pi'$, based on the current policy $\pi$. The key insight is that maximizing this surrogate function, subject to a constraint on the policy change, ensures policy improvement. The theoretical foundation is captured by the following lower bound:

$$J(\pi') \geq J(\pi) + \frac{1}{1-\gamma}\mathbb{E}_{s\sim d_\pi,\, a\sim\pi'(\cdot|s)}\big[A_\pi(s,a)\big]$$
$$- \frac{4\gamma\varepsilon}{(1-\gamma)^2}\max_s D_{\text{KL}}\big(\pi(\cdot\mid s)\,\|\,\pi'(\cdot\mid s)\big). \tag{24}$$

Where $\varepsilon$ represents the bound of the advantage and is defined as follows:

$$|A_\pi(s,a)| \leq \varepsilon, \quad \forall(s,a). \tag{25}$$

TRPO requires data generated by $\pi$ (on policy), making it sample-inefficient. To address these limitations, Off-policy TRPO(Meng et al., 2022) generalizes the surrogate objective by introducing a behavior policy $\mu$. The proposed surrogate function $L_{\pi,\mu}(\tilde{\pi})$ replaces the on-policy state occupancy distribution $d_\pi$ with $d_\mu$:

$$J(\pi') \geq J(\pi) + \frac{1}{1-\gamma}\mathbb{E}_{s\sim d_\mu,\, a\sim\pi(\cdot|s)}\Big[\frac{\pi'(\cdot\mid s)}{\pi(\cdot\mid s)}A_\pi(s,a)\Big]$$
$$- \frac{4\gamma\varepsilon}{(1-\gamma)^2}\Big(\underbrace{D_{\text{KL}}^{\max,sqrt}(\mu,\pi)\,D_{\text{KL}}^{\max,sqrt}(\pi,\pi')}_{\text{off-policy mismatch penalty}} + D_{\text{KL}}^{\max}(\pi,\pi')\Big). \tag{26}$$

Where $L_{\pi,\mu}(\tilde{\pi}) = J(\pi) + \frac{1}{1-\gamma}\mathbb{E}_{s\sim d_\mu,\, a\sim\pi(\cdot|s)}\Big[\frac{\pi'(\cdot|s)}{\pi(\cdot|s)}A_\pi(s,a)\Big]$, $D_{\text{KL}}^{\max}(p,q) = \max_s D_{\text{KL}}\big(p(\cdot\mid s)\,\|\,q(\cdot\mid s)\big)$, and $D_{\text{KL}}^{\max,sqrt}(p,q) = \max_s \sqrt{D_{\text{KL}}\big(p(\cdot\mid s)\,\|\,q(\cdot\mid s)\big)}$. GIPO replaces the raw importance weight $\frac{\pi'(\cdot|s)}{\pi(\cdot|s)}$ by the effective weight $\omega(\bar{\rho}';\sigma)\rho'$, which yields the proposed surrogate described as follows:

$$\tilde{L}(\pi') \triangleq J(\pi) + \frac{1}{1-\gamma}\mathbb{E}_{s\sim d_\mu,\, a\sim\mu(\cdot|s)}\Big[\omega(\bar{\rho}';\sigma)\,\rho'\,A_\pi(s,a)\Big]. \tag{27}$$

## B.2. Proof of Lemma 5.1

Fix a state $s$ and abbreviate $\rho' := \rho'(s,a) = \frac{\pi'(a|s)}{\mu(a|s)}$, where $a\sim\pi'(\cdot\mid s)$. Define

$$Y := |\log\rho'| \geq 0.$$

Recall $\omega(\bar{\rho}';\sigma) = \exp\big(-\frac{Y^2}{2\sigma^2}\big)$, hence

$$1 - \omega(\bar{\rho}';\sigma) = 1 - \exp\Big(-\frac{Y^2}{2\sigma^2}\Big).$$

**Step 1:** For any $u \geq 0$, $1 - e^{-u} \leq \min\{u, 1\}$. With $u := \frac{Y^2}{2\sigma^2}$,

$$1 - \omega(\bar{\rho}'; \sigma) \leq \min\left\{\frac{Y^2}{2\sigma^2}, 1\right\}. \tag{28}$$

Fix any threshold $\tau > 0$. If $Y < \tau$, then $\mathbf{1}\{Y \geq \tau\} = 0$, $\frac{\tau^2}{2\sigma^2} + \mathbf{1}\{Y \geq \tau\} = \frac{\tau^2}{2\sigma^2}$. if $\frac{Y^2}{2\sigma^2} \leq 1$, $\min\left\{\frac{Y^2}{2\sigma^2}, 1\right\} =$ is $\frac{Y^2}{2\sigma^2}$. Thus, $\min\left\{\frac{Y^2}{2\sigma^2}, 1\right\} \leq \frac{\tau^2}{2\sigma^2} + \mathbf{1}\{Y \geq \tau\}$;

if $Y \geq \tau$, the indicator $\mathbf{1}\{Y \geq \tau\} = 1$. Then $\min\left\{\frac{Y^2}{2\sigma^2}, 1\right\} \leq 1 \leq \frac{\tau^2}{2\sigma^2} + 1$

Thus, the following inequality holds pointwise,

$$\min\left\{\frac{Y^2}{2\sigma^2}, 1\right\} \leq \frac{\tau^2}{2\sigma^2} + \mathbf{1}\{Y \geq \tau\}. \tag{29}$$

Taking expectation under $a \sim \pi'(\cdot \mid s)$ and using $\mathbb{E}[\mathbf{1}\{Y \geq \tau\}] = \mathbb{P}(Y \geq \tau)$ gives:

$$\mathbb{E}_{a \sim \pi'(\cdot|s)}[1 - \omega(\bar{\rho}'; \sigma)] \leq \frac{\tau^2}{2\sigma^2} + \mathbb{P}_{a \sim \pi'(\cdot|s)}(|\log \rho'| \geq \tau). \tag{30}$$

**Step 2:** We first bound $\mathbb{P}_{\pi'}(\log \rho' \leq -\tau)$. Note that $\{\log \rho' \leq -\tau\} \iff \{\rho' \leq e^{-\tau}\} \iff \{\frac{1}{\rho'} \geq e^{\tau}\}$. By Markov's inequality,

$$\mathbb{P}_{\pi'}\left(\frac{1}{\rho'} \geq e^{\tau}\right) \leq e^{-\tau} \mathbb{E}_{\pi'}\left[\frac{1}{\rho'}\right]. \tag{31}$$

Moreover,

$$\mathbb{E}_{a \sim \pi'(\cdot|s)}\left[\frac{1}{\rho'}\right] = \sum_a \pi'(a \mid s) \frac{\mu(a \mid s)}{\pi'(a \mid s)} = \sum_a \mu(a \mid s) = 1. \tag{32}$$

Combining (31)–(32) yields

$$\mathbb{P}_{a \sim \pi'(\cdot|s)}(\log \rho' \leq -\tau) \leq e^{-\tau}. \tag{33}$$

**Step 3:** Let $E := \{\log \rho' \geq \tau\} = \{\rho' \geq e^{\tau}\}$. For any event $E$ and two distributions $p, q$ on the same space,

$$\mathbb{P}_p(E) \leq \mathbb{P}_q(E) + D_{\mathrm{TV}}(p, q). \tag{34}$$

Applying (34) with $p = \pi'(\cdot \mid s)$ and $q = \mu(\cdot \mid s)$ gives

$$\mathbb{P}_{\pi'}(\log \rho' \geq \tau) \leq \mathbb{P}_\mu(\log \rho' \geq \tau) + D_{\mathrm{TV}}(\pi'(\cdot \mid s), \mu(\cdot \mid s)). \tag{35}$$

Under $a \sim \mu(\cdot \mid s)$, Markov's inequality and $\mathbb{E}_\mu[\rho'] = 1$ imply

$$\mathbb{P}_\mu(\log \rho' \geq \tau) = \mathbb{P}_\mu(\rho' \geq e^{\tau}) \leq e^{-\tau} \mathbb{E}_\mu[\rho'] = e^{-\tau}, \tag{36}$$

Finally, Pinsker's inequality yields

$$D_{\mathrm{TV}}(\pi'(\cdot \mid s), \mu(\cdot \mid s)) \leq \sqrt{\frac{1}{2} D_{\mathrm{KL}}(\mu(\cdot \mid s) \| \pi'(\cdot \mid s))} \leq \sqrt{\frac{\delta}{2}}. \tag{37}$$

Combining (35)–(37) gives

$$\mathbb{P}_{a \sim \pi'(\cdot|s)}(\log \rho' \geq \tau) \leq e^{-\tau} + \sqrt{\frac{\delta}{2}}. \tag{38}$$

**Step 4:**

$$\mathbb{P}_{\pi'}(|\log \rho'| \geq \tau) = \mathbb{P}_{\pi'}(\log \rho' \geq \tau) + \mathbb{P}_{\pi'}(\log \rho' \leq -\tau) \leq 2e^{-\tau} + \sqrt{\frac{\delta}{2}}. \tag{39}$$

Substituting (39) into (30) yields

$$\mathbb{E}_{a \sim \pi'(\cdot|s)}[1 - \omega(\bar{\rho}'; \sigma)] \leq \frac{\tau^2}{2\sigma^2} + 2e^{-\tau} + \sqrt{\frac{\delta}{2}},$$

which is exactly (17).

### B.3. Optimizing the Truncation parameter $\tau$

The lower bound 5.2 holds for any $\tau > 0$. To obtain the tightest bound, one can minimize the penalty term with respect to $\tau$.

Consider $g(\tau) = \frac{\tau^2}{2\sigma^2} + 2e^{-\tau}$. Setting $g'(\tau) = 0$ yields

$$\frac{\tau}{\sigma^2} = 2e^{-\tau} \quad \Longleftrightarrow \quad \tau e^{\tau} = 2\sigma^2.$$

Thus $\tau^{\star} = W(2\sigma^2)$ ($W$ denotes the Lambert-$W$ function) and $g''(\tau^{\star}) > 0$. Substituting $2e^{-\tau^{\star}} = \tau^{\star}/\sigma^2$ gives

$$g(\tau^{\star}) = \frac{(\tau^{\star})^2}{2\sigma^2} + \frac{\tau^{\star}}{\sigma^2} = \frac{\tau^{\star}(\tau^{\star} + 2)}{2\sigma^2}.$$

# C. Detailed Experimental Settings

## C.1. Target and Advantage Construction

All experiments across the evaluated methods use a shared target and advantage construction based on Generalized Advantage Estimation (GAE). This standardized configuration ensures that any observed performance differences primarily reflect the efficacy of the respective surrogate-side ratio handling strategies, isolating the impact of GIPO's smooth trust weighting.

## C.2. Training Configuration

**General Setup.** To ensure a fair comparison, all methods utilize the same VLA backbone architecture, replay buffer configuration, and mechanisms for value target computation and advantage estimation (GAE), unless explicitly stated otherwise.

**Compute Resources.** Experiments were conducted on a high-performance cluster equipped with NVIDIA H200 GPUs (approx. 1,979 TFLOPS BF16 peak performance per GPU).

- **MetaWorld:** Training employed $1 \times$ H200 GPUs for approximately 10 hours.

- **LIBERO:** Training employed $8 \times$ H200 GPUs for approximately 2 days.

**Hyperparameters.** We use the AdamW optimizer for all experiments with a discount factor $\gamma = 0.99$ and GAE parameter $\lambda = 0.95$. The learning rates are set to $3 \times 10^{-6}$ for the policy and $3 \times 10^{-5}$ for the value function. Specific settings for each environment and algorithm are as follows:

- **Environment Settings:**

  - *MetaWorld*: Batch size $B = 512$, trained for 100,000 iterations.
  - *LIBERO*: Batch size $B = 1024$, trained for 30,000 iterations.

- **Algorithm-Specific Parameters:**

  - **GIPO (Ours):** $\sigma = 1.0$ for MetaWorld and $\sigma = 0.5$ for LIBERO.
  - **PPO:** $\epsilon = 0.2$.
  - **SAPO:** $\tau_{pos} = 2$ and $\tau_{neg} = 1$.

## C.3. Regime quantification and aggregation protocol

This section provides the full regime specification and evidence that the Fresh and Stale settings induce distinct levels of policy lag. Each replay entry stores a behavior-policy version identifier. We define the version gap $\Delta v_t$ as the difference between the current learner policy version and the behavior-policy version associated with sample $t$. We define an old-sample subset using a fixed threshold $T_{old}$ in version-gap units. **For MetaWorld, we set $T_{old} = 10{,}000$ (version-gap units) and use a replay capacity of 50,000.** We summarize staleness using OldFrac and OldGapP95 from Sec. 6.1. We summarize ratio-tail drift using

$$D_{0.95} \triangleq Q_{0.95}(|\log \rho_t|), \qquad \rho_t = \frac{\pi_\theta(a_t \mid o_t)}{\mu(a_t \mid o_t)}. \tag{40}$$

We log $D_{0.95}$ as `AbsLogRho_P95`.

We aggregate regime statistics across tasks using per-task window means over the last 20% of training environment steps. For each task, we compute the mean of each statistic within the final window over logged points. We then report the mean across tasks with the standard deviation across tasks.

Table 2 summarizes the regime configuration and the observed lag and tail statistics. Figure 5 reports representative training-time trajectories of the staleness indicators.

*Table 2.* **Regime quantification.** Configuration and observed statistics. We report the mean across tasks with the standard deviation across tasks using per-task window means over the last 20% of training environment steps.

| Configuration / Statistic | Fresh | Stale |
|---|---|---|
| *Configuration* | | |
| `num_actors` | 16 | 2 |
| Replay capacity | 50,000 | 50,000 |
| Sampling | uniform | uniform |
| *Observed regime statistics, last 20% of env steps* | | |
| $\text{OldFrac} = \Pr(\Delta v \geq T_{\text{old}})$ | $0.000 \pm 0.000$ | $0.884 \pm 0.003$ |
| $\text{OldGapP95} = Q_{0.95}(\Delta v)$ | $0.00 \pm 0.00$ | $8.27 \times 10^4 \pm 9.04 \times 10^2$ |
| $D_{0.95} = Q_{0.95}(|\log \rho|)$ `AbsLogRho_P95` | $0.095 \pm 0.006$ | $1.46 \pm 0.75$ |

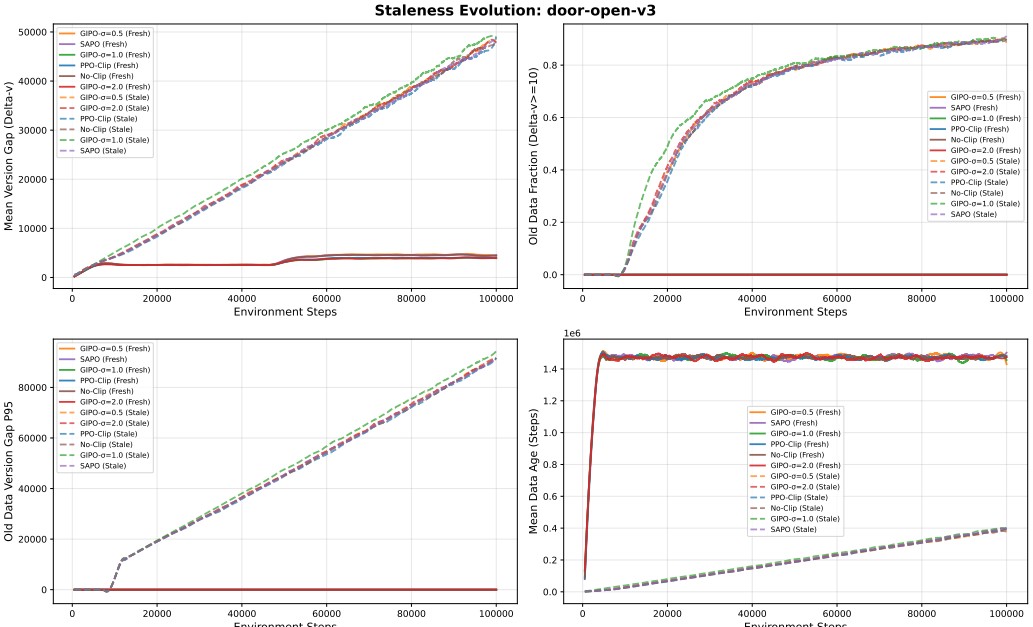

*Figure 5.* **Training-time evolution of staleness indicators on a representative task.** We plot mean version gap, OldFrac, OldGapP95, and mean data age for both regimes. The Stale regime transitions into larger version gaps and higher old-sample fractions, while the Fresh regime remains near the low-lag range throughout training.

## C.4. Utilization metrics

This section defines utilization diagnostics used to interpret how replay samples contribute to the actor update. Let $m_t$ denote the surrogate-induced scalar multiplier on $\nabla_\theta \log \pi_\theta(a_t \mid o_t)$ for sample $t$. Let $\hat{A}_t$ denote the advantage used in the actor update. We use the per-sample contribution proxy

$$u_t \triangleq \left| m_t \, \hat{A}_t \right|. \tag{41}$$

Given thresholds $\tau_u > 0$ and $\tau_m \geq 0$, we define three batch-level fractions

$$\text{NearZeroFrac}(\tau_u) \triangleq \Pr(u_t \leq \tau_u), \tag{42}$$

$$\text{DeadFrac} \triangleq \Pr(|m_t| = 0), \tag{43}$$

$$\text{SuppressedFrac}(\tau_m) \triangleq \Pr(0 < |m_t| \leq \tau_m). \tag{44}$$

Dead samples have $m_t = 0$. Suppressed samples have nonzero but small $|m_t|$. Near-zero samples have negligible contribution magnitude after accounting for $\hat{A}_t$.

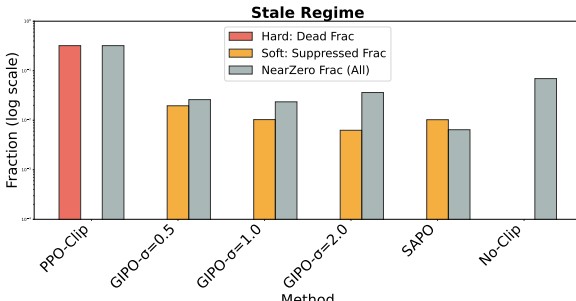

Figure 6. **Hard clipping yields more near-zero contributions under stale replay.** Dead, suppressed, and near-zero contribution fractions on `door-open-v3` in the Stale regime. Fractions are computed from the contribution proxy and thresholds specified in Appendix C.4. The y-axis uses a log scale.

**Old-sample split.** We define the version gap $\Delta v_t$ as in Sec. 6.1. Given a fixed threshold $T_{\text{old}}$, we define the old subset as $\{t : \Delta v_t \geq T_{\text{old}}\}$. The threshold value used in each experiment is part of the regime specification reported in Appendix C.3.

**Old-sample contribution share.** We summarize how much of the total contribution mass comes from old replay by

$$\text{Share}_{\text{old}} \triangleq \frac{\sum_{t \in \text{old}} u_t}{\sum_t u_t}. \tag{45}$$

**Effective sample size on old samples.** To summarize concentration of contribution weights within the old subset, we compute effective sample size from normalized contribution weights. Let $w_t = u_t$ for $t$ in the old subset and define normalized weights $\tilde{w}_t = w_t / \sum_{j \in \text{old}} w_j$. We define

$$\text{ESS}_{\text{eff,old}} \triangleq \frac{1}{\sum_{t \in \text{old}} \tilde{w}_t^2}, \tag{46}$$

and report a normalized variant

$$\text{ESS}_{\text{eff,old}}^{\text{norm}} \triangleq \frac{\text{ESS}_{\text{eff,old}}}{|\text{old}|}. \tag{47}$$

**Windowed reporting.** When used as scalars in summary figures, these metrics are reported as window means over the last portion of training, using the same aggregation protocol as in Sec. 6.1.

### C.5. MetaWorld diagnostics under stale replay

This section reports additional diagnostics on the representative task `door-open-v3`. The goal is to relate stale replay to how replay samples contribute to the actor update, and to provide a compact view of the association between update size, old-data utilization, and learning outcomes.

**Near-zero contributions under hard clipping.** Figure 6 reports dead, suppressed, and near-zero contribution fractions in the Stale regime. Fractions are computed from the contribution proxy in Eq. (41) using the thresholds specified in Appendix C.4. In the reported run, **PPO-Clip** has a larger near-zero fraction than the smooth-weighting methods. This indicates that some replayed samples contribute negligibly to the actor update once ratios fall outside the clipping region.

**Stability–utilization diagnostic in the stale regime.** Figure 7 summarizes each configuration on a common plane for `door-open-v3` in the Stale regime. KL divergence serves as a coarse proxy for update size. Normalized old-data ESS $\text{ESS}_{\text{eff,old}}^{\text{norm}}$ measures effective utilization of stale replay. Color indicates evaluation average return. Each point is a window mean over the last 20% of training environment steps for one configuration.

In the reported run, configurations occupy different regions of this plane. **No-Clip** reaches relatively large KL while $\text{ESS}_{\text{eff,old}}^{\text{norm}}$ is near zero and return is low. **PPO-Clip** remains at low KL but also attains lower returns than several smooth-

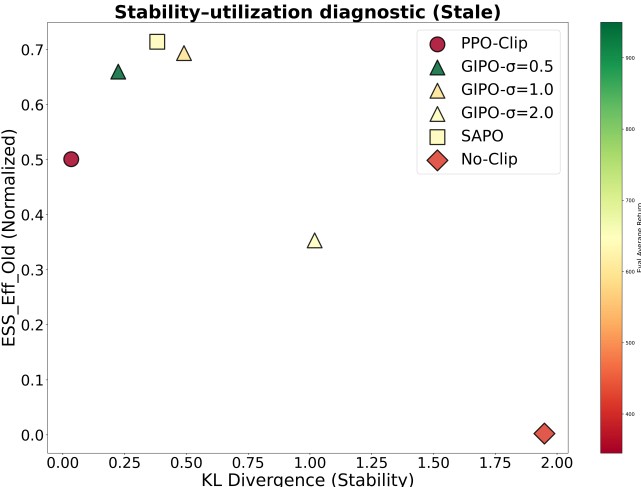

*Figure 7.* **Stability–utilization diagnostic under stale replay on `door-open-v3`.** KL divergence versus normalized old-data ESS $\mathrm{ESS}^{\mathrm{norm}}_{\mathrm{eff,old}}$, with color indicating evaluation average return. Points are window means over the last 20% of training environment steps.

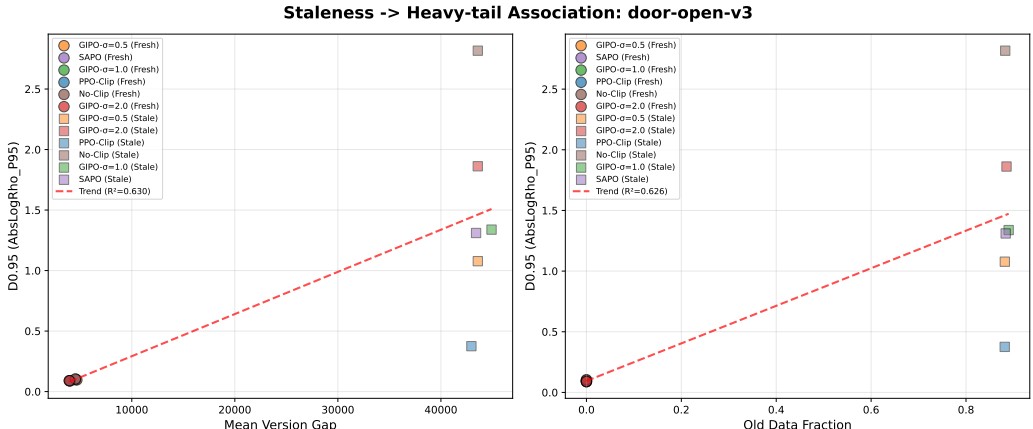

*Figure 8.* **Policy lag versus ratio-tail drift on a representative task.** We plot $D_{0.95} = Q_{0.95}(|\log \rho|)$ against staleness summaries on `door-open-v3`. The fitted trend line and $R^2$ are reported for reference.

weighting configurations in the same run. This diagnostic provides a descriptive view that connects outcome differences to utilization of old replay.

### C.6. Lag and ratio-tail association

This section provides a descriptive view of how policy lag relates to importance-ratio tail behavior in the replay-heavy setting. We quantify lag using the staleness summaries derived from $\Delta v_t$ and summarize tail behavior using $D_{0.95} = Q_{0.95}(|\log \rho|)$. Figure 8 plots $D_{0.95}$ against lag summaries on `door-open-v3`. This plot is descriptive and does not establish causality.

### C.7. Sensitivity to the damping scale $\sigma$

We report a diagnostic ablation on another task, `handle-press-v3` to illustrate how the GIPO damping scale $\sigma$ relates to ratio-tail drift, old-data utilization, and evaluation return in the reported runs. We evaluate $\sigma \in \{0.5, 1, 2\}$ under both Fresh and Stale regimes. Figure 9 reports $D_{0.95}$, `ESS_Eff_Old`, and evaluation average return for each setting. This study is task-specific and is included as descriptive evidence.

**GIPO Sigma Sensitivity: handle-press-v3**

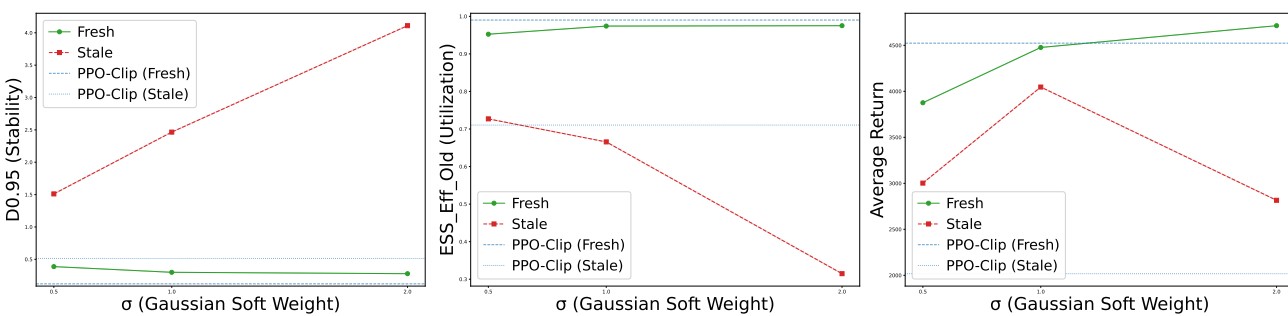

*Figure 9.* **Sensitivity to the GIPO damping scale on a representative task.** Results on `handle-press-v3` for $\sigma \in \{0.5, 1, 2\}$ under Fresh and Stale regimes. We report $D_{0.95}$, `ESS_Eff_Old`, and evaluation average return.

## C.8. Toy Experiment Setting

The environment Grid 2 * 2 is defined as follows:

- **State space:** $\mathcal{S} = \{S_0, S_1, S_2, S_G\}$, with a fixed initial state $S_0$ and a terminal absorbing goal state $S_G$. Illustration of gridworld is shown in Figure 4.

- **Action space:** $\mathcal{A} = \{\text{up}, \text{down}, \text{left}, \text{right}\}$. Transitions are deterministic. When the agent takes an action, it moves in the corresponding direction; if the action would leave the grid, the state remains unchanged.

- **Reward:** $r(s_t, a_t, s_{t+1}) = -1$. After reaching $S_G$, the process remains absorbing and no further reward is received.

**Behavior policies and cases.** In this experiment, all policies are parameterized as softmax distributions over actions (ordered as $[\text{up}, \text{down}, \text{left}, \text{right}]$):

$$\pi(a \mid s) = \frac{\exp(\theta_a)}{\sum_{a'} \exp(\theta_{a'})}. \tag{48}$$

For any non-terminal state, policy $\pi$ is fixed as $\theta^\pi = [0, 1, 0, 1]$. Specially, the policy is as follow:

$$
\begin{aligned}
\pi(\text{up} \mid s) = \pi(\text{left} \mid s) &= \frac{1}{2(1+e)} \approx 0.1345, \\
\pi(\text{down} \mid s) = \pi(\text{right} \mid s) &= \frac{e}{2(1+e)} \approx 0.3655.
\end{aligned}
\tag{49}
$$

To simulate different degrees of policy lag, we define three behavior policies $\mu$ by combining three base behavior policies:

- random policy: $\theta^{\text{rand}} = [0, 0, 0, 0]$

- right-preferring policy: $\theta^{\text{right}} = [0, 0, 0, 1]$

- down-preferring policy: $\theta^{\text{down}} = [0, 1, 0, 0]$

The three behavior policies $\mu$ are defined as:

- **Case A:** $\mu = \pi^{\text{rand}}$

- **Case B:** $\mu = 0.4\,\pi^{\text{rand}} + 0.3\,\pi^{\text{right}} + 0.3\,\pi^{\text{down}}$

- **Case C:** $\mu = 0.2\,\pi^{\text{rand}} + 0.4\,\pi^{\text{right}} + 0.4\,\pi^{\text{down}}$

From Case A to Case C, $\mu$ assigns progressively more probability mass to down/right and becomes increasingly similar to $\pi$, i.e., policy lag decreases.

**Metrics.** For each method, we treat the one-step gradient estimator as a random variable $g(a)$ under $a \sim \mu(\cdot \mid s)$, and compute exactly by action enumeration:

$$\mathbb{E}_\mu[g] = \sum_a \mu(a \mid s)g(a),$$

$$\mathrm{Var}_\mu[g] = \sum_a \mu(a \mid s)\big(g(a) - \mathbb{E}_\mu[g]\big)^2. \tag{50}$$

We define bias as the deviation of $\mathbb{E}_\mu[g]$ from the ground-truth gradient, and variance as $\mathrm{Var}_\mu[g]$.

# D. Additional Comprehensive Multi-Seed Evaluations

To validate the algorithmic universality of GIPO and ensure statistical rigor across diverse environmental complexities, we conducted an extensive suite of evaluations on both Classic Control and MuJoCo benchmarks. Across all following experiments, each configuration is evaluated using 5 random seeds. In the following experiments, Standard GIPO is denoted by GIPO ($\sigma$), while the Advantage-Aware version is denoted by GIPO ($\sigma_{\text{pos}}, \sigma_{\text{neg}}$).

## D.1. Classic Control Benchmarks

We first evaluate our method on four standard Gym Classic Control tasks to test its efficacy in both discrete and continuous action spaces. As shown in Table 3 and Table 4, GIPO demonstrates superior convergence speed and higher final returns compared to PPO and SAPO. These results confirm that GIPO's smooth gradient contributions effectively prevent utilization collapse under policy lag, maintaining robust training stability and improving sample efficiency even with highly stale replay data.

*Table 3.* Final training rollout average returns on Classic Control tasks (mean $\pm$ standard deviation over 5 seeds).

| TASK | PPO | SAPO | GIPO (0.2) | GIPO (0.5) | GIPO (1.0) |
|---|---|---|---|---|---|
| CARTPOLE-V1 | $333.29 \pm 94.65$ | $359.08 \pm 119.76$ | $\mathbf{449.15 \pm 39.70}$ | $391.95 \pm 92.86$ | $394.11 \pm 92.99$ |
| ACROBOT-V1 | $-92.10 \pm 2.34$ | $-90.64 \pm 2.58$ | $-93.79 \pm 3.51$ | $-90.67 \pm 2.02$ | $\mathbf{-89.66 \pm 3.74}$ |
| MOUNTAINCARCONT. | $-40.51 \pm 3.32$ | $-41.23 \pm 1.90$ | $\mathbf{-37.71 \pm 7.57}$ | $-40.29 \pm 3.78$ | $-39.26 \pm 5.80$ |
| PENDULUM-V1 | $-215.0 \pm 65.1$ | $-226.2 \pm 54.0$ | $-276.8 \pm 193.0$ | $\mathbf{-177.3 \pm 6.6}$ | $-199.6 \pm 42.2$ |

*Table 4.* Aggregate IQM of Normalized Scores across the four Classic Control tasks.

| RANK | ALGORITHM | IQM SCORE |
|---|---|---|
| 1 | GIPO (0.2) | 0.8095 |
| 2 | GIPO (1.0) | 0.8065 |
| 3 | GIPO (0.5) | 0.7988 |
| 4 | PPO | 0.7908 |
| 5 | SAPO | 0.7495 |

## D.2. MuJoCo Continuous Control Benchmarks

We next evaluate GIPO on 10 high-dimensional MuJoCo tasks to assess its robustness against policy lag in continuous control settings. As reported in Table 5, GIPO's standard symmetric variants rank among the top performers compared to both PPO and SAPO. To eliminate outlier variance, we follow the RLiable framework to compute the Interquartile Mean (IQM) of Normalized Scores. As shown in Table 7, GIPO variants securely occupy the top rankings, with GIPO (0.5) claiming the highest score of 0.606.

*Table 5.* Multi-seed Average Return (Mean $\pm$ Std) on the 10 MuJoCo benchmarks over 5 random seeds, comparing standard symmetric GIPO against PPO and SAPO baselines.

| TASK | PPO | SAPO | GIPO (0.2) | GIPO (0.5) | GIPO (1.0) |
|---|---|---|---|---|---|
| HALFCHEETAH | $-1158.6 \pm 893.2$ | $869.1 \pm 324.5$ | $1718.1 \pm 699.3$ | $\mathbf{2867.0 \pm 1255.5}$ | $2426.4 \pm 1084.9$ |
| ANT | $-2261.7 \pm 1277.0$ | $23.6 \pm 10.0$ | $\mathbf{977.1 \pm 359.8}$ | $822.9 \pm 707.2$ | $20.4 \pm 20.5$ |
| HOPPER | $1.7 \pm 2.8$ | $455.4 \pm 394.7$ | $786.9 \pm 717.6$ | $870.8 \pm 280.1$ | $346.2 \pm 224.0$ |
| WALKER2D | $-24.0 \pm 0.7$ | $600.7 \pm 458.8$ | $1510.9 \pm 1250.9$ | $\mathbf{1610.3 \pm 1661.0}$ | $793.5 \pm 401.1$ |
| HUMANOID | $61.7 \pm 4.8$ | $635.4 \pm 145.1$ | $\mathbf{789.7 \pm 258.2}$ | $716.4 \pm 148.1$ | $549.9 \pm 81.2$ |
| INVERTEDPENDULUM | $3.4 \pm 0.9$ | $408.2 \pm 540.4$ | $\mathbf{808.4 \pm 428.4}$ | $805.2 \pm 435.6$ | $610.2 \pm 534.0$ |
| INVDOUBLEPENDULUM | $32.0 \pm 4.6$ | $7498.7 \pm 4125.6$ | $7497.0 \pm 4115.0$ | $\mathbf{7546.0 \pm 4016.1}$ | $5632.8 \pm 5092.6$ |
| REACHER | $-103.8 \pm 1.9$ | $-48.2 \pm 37.8$ | $\mathbf{-16.4 \pm 21.3}$ | $-50.7 \pm 40.2$ | $-42.7 \pm 35.6$ |
| PUSHER | $-314.6 \pm 49.9$ | $-107.2 \pm 35.6$ | $-66.0 \pm 13.7$ | $\mathbf{-62.7 \pm 10.2}$ | $-79.4 \pm 28.7$ |
| HUMANOIDSTANDUP | $55137.2 \pm 9866.9$ | $112159.4 \pm 23746.0$ | $\mathbf{142419.5 \pm 13901.5}$ | $118900.8 \pm 26565.1$ | $98645.0 \pm 7688.7$ |

*Table 6.* Ablation study on three core MuJoCo tasks comparing standard GIPO against the Advantage-Aware GIPO variants. All variants are denoted by $(\sigma_{\text{pos}}, \sigma_{\text{neg}})$.

| TASK | GIPO (0.2, 0.2) | GIPO (0.5, 0.5) | GIPO (1.0, 1.0) | ADV-AWARE (0.2, 0.1) | ADV-AWARE (0.5, 0.25) | ADV-AWARE (1.0, 0.5) |
|---|---|---|---|---|---|---|
| HALFCHEETAH | $1718.1 \pm 699.3$ | $\mathbf{2867.0 \pm 1255.5}$ | $2426.4 \pm 1084.9$ | $1848.8 \pm 1005.9$ | $1870.5 \pm 1002.6$ | $1740.4 \pm 940.1$ |
| ANT | $\mathbf{977.1 \pm 359.8}$ | $822.9 \pm 707.2$ | $20.4 \pm 20.5$ | $819.8 \pm 521.5$ | $367.2 \pm 815.9$ | $27.4 \pm 13.8$ |
| HOPPER | $786.9 \pm 717.6$ | $870.8 \pm 280.1$ | $346.2 \pm 224.0$ | $\mathbf{1184.1 \pm 673.2}$ | $1106.6 \pm 1162.5$ | $1156.5 \pm 806.3$ |

## D.3. Ablation Study: Advantage-Aware GIPO

To address the potential limitation of advantage-sign-agnostic damping, we evaluate an Advantage-Aware GIPO variant that applies different trust region scales based on the advantage sign. As shown in Table 6, we present a targeted ablation study on three core tasks. The results demonstrate that while the standard GIPO configurations (e.g., GIPO (0.5, 0.5)) remain highly robust as a default baseline, the Advantage-Aware counterparts (e.g., GIPO (0.2, 0.1) on Hopper) can successfully extract further performance gains in specific environments. This validates that GIPO seamlessly accommodates applying different $\sigma$ scales for positive and negative advantages while preserving its core log-space symmetry and training stability.

*Table 7.* Aggregate IQM of normalized scores across MuJoCo tasks for Advantage-Aware GIPO variants, denoted by $(\sigma_{\text{pos}}, \sigma_{\text{neg}})$.

| RANK | ALGORITHM | IQM SCORE |
|---|---|---|
| 1 | **GIPO (0.5, 0.5)** | **0.606** |
| 2 | GIPO (0.5, 0.25) | 0.602 |
| 3 | GIPO (0.2, 0.1) | 0.570 |
| 4 | GIPO (0.2, 0.2) | 0.551 |
| 5 | GIPO (1.0, 0.5) | 0.533 |
| 6 | GIPO (1.0, 1.0) | 0.478 |
| 7 | SAPO | 0.413 |
| 8 | PPO | 0.119 |

