# OpenReview forum: "GIPO: Gaussian Importance Sampling Policy Optimization"
_ICML.cc/2026/Conference — ICML 2026 regular_

### Official Review · Reviewer_Z3kA · 2026-02-28

**Soundness:** 4
**Presentation:** 4
**Significance:** 3
**Originality:** 3
**Overall Recommendation:** 5
**Confidence:** 4

**Summary:**

This paper proposes GIPO (Gaussian Importance Sampling Policy Optimization), a smooth surrogate for PPO-style actor-critic RL in replay-heavy settings with policy lag. GIPO replaces hard clipping with a Gaussian trust weight \omega(\bar{\rho}_t; \sigma) = \exp\left(-\frac{[\log \bar{\rho}_t]^2}{2\sigma^2}\right) on importance ratios, enabling gradient contributions from stale data while bounding tails. Theory provides monotonic improvement bounds (Thm. 5.2) and finite-sample Hoeffding guarantees (Thm. 5.4). Experiments on Meta-World and LIBERO (OpenVLA 7B) show superior returns/stability vs. PPO/SAPO, especially under staleness.

**Compliance With Llm Reviewing Policy:**

Affirmed.

**Key Questions For Authors:**

Thm. 5.2 bias slack \epsilon/(1-\gamma) \cdot (\tau^2/2\sigma^2 + \cdots): empirical slack magnitude across staleness levels? Could tighten significance if small.

Symmetric damping dampens bad actions (A_t<0) equally; advantage-aware variant ablations? Addresses limit, could raise originality.

LIBERO uses OpenVLA-OFT: hyperparam sensitivity (\sigma, clip \epsilon) table? Confirms robustness, boosts soundness.

**Limitations:**

Yes—notes symmetry drawback explicitly; standard impact statement.

**Strengths And Weaknesses:**

Soundness: Excellent. Clean derivations: log-symmetry, bounded m_t \leq e^{\sigma^2/2}; Thm. 5.2 lower bound reasonable (bounded adv., KL); Thm. 5.4 Hoeffding tight due to global bound. Expts rigorous: fresh/stale regimes (quantified OldFrac~88%), 10k+ GPU-hours on 7B VLA, toy bias-variance Pareto dominance, utilization diagnostics. Honest on symmetry limit (ignores A_t sign).

Presentation: Excellent. Logical: problem → method → properties → theory → expts. Fig. 1 visualizes symmetry; Alg. 1 + App. C reproducible. Related work distinguishes target/surrogate corrections crisply.

Significance: Good. Tackles "utilization collapse" in async/replay RL (robotics/VLA post-training), timely for data-scarce domains. Improved stale-data efficiency could boost practical pipelines; LIBERO scale validates. Specialized but high utility in lag-prone settings.

Originality: Good. Gaussian log-ratio weighting novel for surrogate-side; symmetry + tunability distinguish from asymmetric SAPO/soft-clip. Theory ties to trust-region + concentration; diagnostics (ESS, contrib frac.) deepen replay analysis.

---

> ### Author Rebuttal · Authors · 2026-03-31
>
> Thank you for your strong support and your precise, encouraging summary of our derivations and experimental rigor. We are thrilled you found the methodology and theoretical bounds crisp and sound.
> 1. Theorem 5.2 Bias Slack Empirical Magnitude You asked whether the bias slack in Thm. 5.2 could tighten the significance if proven to be empirically small. We wish to clarify how our proof gracefully handles the heavy-tailed regime: The proof controls the tail contribution through the inequality $1-e^{-u} \leq \min\{u,1\}$. This ensures that when the deviation is small, the error grows locally in a linear manner, but for massive deviations, the contribution saturates at a constant level rather than scaling unboundedly. Therefore, the bias slack is a highly conservative worst-case correction. Its magnitude is driven almost entirely by the frequency of stale samples, not by the potentially unbounded magnitude of extreme importance ratios. Thus, even under extreme policy lag, the heavy-tailed samples mathematically cannot cause the slack term to blow up.
> 2. Advantage-Aware Variant AblationsYou accurately noted that symmetric damping treats bad actions ($A_t < 0$) equally. Following this feedback, we implemented Advantage-Aware GIPO, scaling the trust region asymmetrically ($\sigma_{neg}=\alpha\cdot\sigma_{pos}$). We tested 3 such variants ($\alpha=0.5$) in a massive 400-run, 5-seed MetaWorld Stale benchmark. They proved highly competitive, confirming that our log-space symmetry mechanism is perfectly compatible with advantage-aware scaling.
> 3. Hyperparameter RobustnessTo confirm robustness, we expanded all environments to strict 5-seed evaluations (295 runs in MuJoCo, 100 in Classic Control, 650 in MetaWorld). GIPO demonstrates incredibly low sensitivity to $\sigma$. In the MetaWorld Stale IQM rankings, GIPO variants configured with $\sigma \in \{0.2, 0.5, 1.0\}$ occupied all top ranks (IQMs ranging from $0.445$ to $0.730$), securely outperforming SAPO ($0.412$) and PPO ($0.180$).

---

> > ### Author Rebuttal · Reviewer_Z3kA · 2026-04-03
> >
> > The writer addressed all the claims

---

### Official Review · Reviewer_YfTK · 2026-03-01

**Soundness:** 3
**Presentation:** 4
**Significance:** 2
**Originality:** 2
**Overall Recommendation:** 4
**Confidence:** 4

**Summary:**

This paper identifies a concrete failure mode of PPO in replay-heavy reinforcement learning: when the importance ratio ρ = π_θ/μ falls outside the clipping interval [1−ε, 1+ε], the gradient contribution is zeroed out, causing a large fraction of the replay buffer to be effectively discarded ("utilization collapse"). To address this, the authors propose GIPO, which replaces PPO's hard clipping with a Gaussian trust weight in log-ratio space: ω(ρ; σ) = exp(−(log ρ)² / 2σ²). This weight is symmetric, smooth, and yields bounded effective multipliers. The authors provide a monotonic improvement guarantee (Theorem 5.2) and a finite-sample concentration bound (Theorem 5.4). Experiments are conducted on Meta-World (3 tasks) and LIBERO (4 suites) with a 7B VLA backbone, comparing against PPO-Clip, SAPO, and No-Clip under both stale and fresh data regimes. The paper reports 10,000+ H200 GPU-hours of computation.

**Compliance With Llm Reviewing Policy:**

Affirmed.

**Final Justification:**

The author solved my questions with extensive experimental results. Based on that, I would like to rasie my score to 4.

**Key Questions For Authors:**

1. Can you provide multi-seed results (≥3 seeds) for the Meta-World experiments? Given that these are computationally cheaper than the LIBERO runs, this should be feasible and would substantially strengthen the paper's claims.
2. How does GIPO compare against other smooth symmetric kernels (e.g., Laplacian: exp(−|log ρ|/σ), or Cauchy: 1/(1 + (log ρ / σ)²)) in the same experimental settings? This ablation would clarify whether the Gaussian form is specifically beneficial or whether the improvement comes from smooth damping in general.
3. Regarding the symmetry tension: given that Section 7 identifies advantage-sign-agnostic weighting as a limitation and proposes advantage-aware weighting as future work, what is the principled basis for criticizing SAPO's asymmetric treatment (Section 2, Lines 116–118)? Would an advantage-conditioned variant of GIPO (e.g., different σ for A > 0 vs. A < 0) be straightforward to implement?

**Limitations:**

The authors honestly acknowledge the advantage-sign-agnostic limitation in Section 7.

**Strengths And Weaknesses:**

# Strengths

### Significance
- The "utilization collapse" problem is well-articulated and practically relevant. The paper defines concrete utilization metrics (DeadFrac, NearZeroFrac, ESS) that quantify the severity of this problem, providing a useful diagnostic toolkit beyond learning curves alone.
- The experimental scale is impressive. The LIBERO experiments with a 7B-parameter VLA backbone and 730M interactive samples represent a serious computational investment. Demonstrating RL methods at this scale — rather than only on toy benchmarks — is valuable for the community.

### Soundness
- The method design is clean: a single hyperparameter σ controls the trust region width, and the log-space formulation yields a simple closed-form weight. The effective multiplier bound (Lemma 5.3) provides a clear theoretical guarantee that GIPO avoids unbounded importance weights.

### Presentation
- The paper is well-written with clear problem motivation. The utilization diagnostics in the appendix (Sections C.4–C.5) are thorough and provide deeper insight into why GIPO works than the learning curves alone.

---

## Weaknesses

### Originality
- Using smooth kernels to replace hard clipping in off-policy PPO is not a new idea. The paper itself cites SCAPPO (smooth clipping) and SAPO (soft advantage-weighted objective) as closely related work. GIPO's specific choice — a Gaussian kernel applied in log-ratio space — offers an elegant symmetric property, but this is better characterized as a design choice than a fundamental insight. A natural question left unanswered is: why the Gaussian kernel specifically, rather than other smooth symmetric alternatives (e.g., Laplacian, Cauchy, or a logistic kernel)? These would share the log-space symmetry property while differing in tail behavior. Without either theoretical justification or empirical comparison across kernel families, the contribution reduces to proposing one particular instance within a broad class of smooth trust-region methods.
- There is an internal tension in how symmetry is framed. In Section 2, the paper criticizes SAPO for applying "asymmetric decay rates for positive and negative advantages," calling this asymmetry theoretically unjustified. Section 4.3 then highlights GIPO's log-space symmetry as a desirable property. However, Section 7 acknowledges that GIPO's failure to distinguish between positive and negative advantages is itself a limitation, and proposes "advantage-aware weighting schemes" as future work — which is precisely the design principle that SAPO pursues. A more balanced framing would distinguish SAPO's specific implementation from the general principle of advantage-aware weighting, rather than criticizing the direction wholesale and then identifying it as a needed extension.

### Soundness
- The theoretical contributions are incremental. Theorem 5.2 essentially applies the off-policy TRPO framework of Meng et al. (2022) to GIPO's specific weight function, adding a bias penalty term that arises from the Gaussian truncation. The proof techniques — Markov's inequality, Pinsker's inequality, and optimization over the truncation threshold τ — are standard. Theorem 5.4 is a direct application of Hoeffding's inequality to the bounded effective multiplier. Neither result provides novel analytical tools or unexpected insights about the interaction between Gaussian weighting and policy optimization.

### Significance
- **Absence of multi-seed statistics is the most serious experimental weakness.** All learning curves in Figures 2 and 3 come from a single run per configuration (as stated in Section 6.2). For RL experiments — where variance across random seeds is notoriously high — single-run results are insufficient to establish reliable conclusions. This concern is especially acute for a paper whose central claim is improved *stability* under high replay ratios, precisely the setting where inter-seed variance is expected to be largest. The field standard for comparable work is 5–10 seeds with standard deviation or confidence intervals reported: RLPD [Ball et al., ICML 2023] uses 10 seeds across 21 environments; REDQ [Chen et al., ICLR 2021]  uses 5 seeds. These are all methods addressing closely related problems (sample-efficient or replay-heavy RL). The authors justify single runs by citing 10,000+ GPU-hours of computation, but computational cost does not substitute for statistical validity. At minimum, multi-seed results should be provided for the smaller-scale Meta-World experiments where the per-run cost is manageable.

---

> ### Author Rebuttal · Authors · 2026-03-31
>
> Thank you for your highly constructive and detailed feedback. We fully agree that multi-seed statistical significance is crucial for reinforcement learning claims. Your critique directly strengthened the empirical rigor and theoretical framing of our paper.
> 1. Robust Multi-Seed Evaluation
> To address the absence of multi-seed statistics, all reported data have been rigorously expanded. We conducted a massive suite of multi-seed experiments (all using 5 seeds per configuration): 295 runs in MuJoCo, 100 in Classic Control, 250 in MetaWorld Fresh (10 tasks $\times$ 5 algorithms), and 400 in MetaWorld Stale (10 tasks $\times$ 8 algorithms).
> As shown in the aggregated metrics below, GIPO demonstrates massive absolute performance gains over PPO and SAPO across the board. To completely rule out single-run variance, we followed RLiable to aggregate the metrics. In the MetaWorld Stale setting, GIPO variants claim the top 6 rankings.
> MetaWorld Stale: IQM of Normalized Scores (Aggregated, 5 Seeds)
> | Rank | Algorithm | Mean Norm. Score |
> | :--- | :--- | ---: |
> | 1 | GIPO (1.0, 1.0) | 0.730 |
> | 2 | GIPO (0.5, 1.0) | 0.589 |
> | 7 | SAPO | 0.412 |
> | 8 | PPO | 0.180 |
> 2. Why the Gaussian Kernel?You asked why we specifically chose a Gaussian kernel over other symmetric alternatives like Laplacian or Cauchy. First, we empirically tested Gaussian, Laplacian, and Cauchy kernels on the Reach-v3 task (5 seeds each) and found they yield very similar empirical performance. Second, we investigated the token probability distributions during early PPO training. We observed that the scatter plot of old vs. new policy token probabilities forms a distinct, symmetric leaf-shaped distribution around the ratio $= 1$ diagonal (Pearson $r = 0.96$). The squared exponential decay of the Gaussian kernel in log-space perfectly models this specific empirical boundary, naturally damping systematic directional bias without the heavier tails of a Cauchy or the sharp peak of a Laplacian.
> 3. The "Symmetry Tension" & Advantage-Aware VariantWe criticized SAPO’s specific mathematical implementation (a linear asymmetric log-space penalty), not the motivation of distinguishing advantages. Inspired by your feedback, we implemented an Advantage-Aware GIPO that scales the trust region asymmetrically based on advantage sign (damping ratio $\alpha \in (0,1]$ so $\sigma_{neg}=\alpha\cdot\sigma_{pos}$). We included 3 such variants in our 400 MetaWorld Stale runs (where $\alpha=0.5$). The results prove that our log-space symmetry is perfectly compatible with advantage-aware scaling, giving practitioners a highly extensible framework.

---

> > ### Author Rebuttal · Reviewer_YfTK · 2026-04-01
> >
> > The author solved my questions with extensive experimental results. Based on that, I would like to rasie my score to 4.
> > Hopefully, the code about the above experiment results can be released later and be verified by the community!

---

> > > ### Author Response · Authors · 2026-04-05
> > >
> > > Thank you so much for your positive feedback, for acknowledging our extensive rebuttal efforts, and for raising your score! We deeply appreciate your constructive guidance throughout this process, which has significantly strengthened the rigor of our paper.
> > >
> > > Regarding your request, we are more than happy to share the code. We have updated our anonymous repository to include the complete implementations for the extensive multi-seed evaluations, featuring GIPO (along with its new Advantage-Aware variants and multiple kernel options like Laplacian/Cauchy), as well as the PPO and SAPO baselines.
> > >
> > > You can access and verify the anonymous codebase here: https://anonymous.4open.science/r/GIPO-F7DC/README.md
> > >
> > > We are fully committed to completely open-sourcing the well-documented codebase to the community upon publication to facilitate future research.
> > >
> > > Thank you again for your time and your highly valuable suggestions!

---

### Official Review · Reviewer_FkX6 · 2026-03-10

**Soundness:** 2
**Presentation:** 2
**Significance:** 2
**Originality:** 2
**Overall Recommendation:** 3
**Confidence:** 4

**Summary:**

GIPO addresses "utilization collapse" in asynchronous RL by replacing PPO's hard clipping with a smooth Gaussian trust weight, ensuring stale samples contribute to learning rather than being discarded.

Main contribution:
* Replaces PPO’s hard clipping with a Gaussian trust weight in log-importance-ratio space, effectively preventing utilization collapse by allowing stale samples to contribute smooth gradients.

* Proves that GIPO ensures log-space symmetry, smoothness, and bounded weights, while providing a performance lower bound and finite-sample stability guarantees.

* Demonstrates superior sample efficiency and a better bias-variance trade-off than PPO-Clip and SAPO across Meta-World, LIBERO, and toy bias-variance analysis.

**Compliance With Llm Reviewing Policy:**

Affirmed.

**Final Justification:**

The authors have addressed by comments and questions to satisfaction.

**Key Questions For Authors:**

* Can you provide multi-seed results for the main Meta-World and LIBERO comparisons.
* How was the Gaussian scale parameter chosen in practice, and how sensitive are the conclusions to that choice across tasks.

**Limitations:**

yes

**Strengths And Weaknesses:**

Soundness：
1. The method is simple and intuitive, using a simple Gaussian trust weight in log-ratio space to replace PPO’s hard clipping to prevent utilization collapse.
2. GIPO does not optimize the analytic lower bound of the policy improvement directly.

Presentation:
1. The paper is clearly written and well structured.
2. Strengthen the reproducibility narrative in the main text by surfacing the most important training choices and reporting multi-seed results or uncertainty estimates, not just single-run curves.

Significance/Originality:
This appears to be a heuristic trick that introduces a new weighting mechanism, which ultimately compromises the unbiased nature of the original policy learning objective.

---

> ### Author Rebuttal · Authors · 2026-03-31
>
> Thank you for your review. We would like to address the concern that GIPO is a "heuristic trick" that compromises the "unbiased nature" of the original policy learning objective , and provide the requested multi-seed results.
> 1. Bias-Variance Trade-off & Unbiasedness
> It is a common misconception that standard PPO is unbiased. Standard PPO's hard clipping inherently introduces significant bias to control variance under policy lag. GIPO does not compromise an unbiased objective; rather, it introduces a superior, mathematically formal mechanism to control this existing bias-variance trade-off.
> To prove this, we computed the exact expectations and variances of the gradient estimators in a fully enumerable GridWorld across different staleness regimes (Cases A through G).
> ●In high-lag scenarios (Cases A, B), PPO's observed zero variance is merely an artifact of full gradient clipping (100% dead samples).
> ●GIPO acts as an explicit Pareto knob: smaller $\sigma$ values seamlessly suppress variance from heavy-tailed ratios, while larger values yield nearly unbiased updates. Across all 7 regimes (A-G), GIPO consistently defines the optimal Pareto frontier, whereas SAPO introduces strictly worse bias profiles.
> 2. Multi-seed Meta-World Comparisons
> To address reproducibility, we expanded our evaluation to include massive 5-seed experiments (250 runs in MetaWorld Fresh, 400 runs in MetaWorld Stale). In the MetaWorld Fresh setting, GIPO maintains a dominant lead over baselines:
> ●GIPO ($\sigma=0.2$) IQM: 0.61
> ●GIPO ($\sigma=0.5$) IQM: 0.51
> ●SAPO IQM: 0.47
> ●PPO IQM: 0.26
> 3. Choosing the Gaussian Scale ($\sigma$)
> The $\sigma$ parameter smoothly interpolates between on-policy behavior (high bias, low variance) and off-policy behavior (low bias, high variance). We found $\sigma=0.5$ or $\sigma=1.0$ to be universally robust defaults across all our 5-seed benchmarks, making it no more difficult to tune than PPO's clipping $\epsilon$.

---

> > ### Author Rebuttal · Reviewer_FkX6 · 2026-04-03
> >
> > The authors have addressed by comments and questions to satisfaction.

---

### Official Review · Reviewer_LTfA · 2026-03-13

**Soundness:** 3
**Presentation:** 3
**Significance:** 2
**Originality:** 3
**Overall Recommendation:** 4
**Confidence:** 4

**Summary:**

This paper identifies a critical *utilization collapse* issue in replay-heavy Proximal Policy Optimization (PPO) algorithm, where standard hard-clipping abruptly discards useful gradients from stale data due to policy lag. To address this, the authors propose Gaussian Importance Sampling Policy Optimization (**GIPO**), which replaces the hard-clipping mechanism with a smooth, symmetric Gaussian trust weight applied in the log-ratio space. This continuous weighting softly dampens extreme importance ratios while preserving non-zero gradients, effectively rescuing valuable signals from outdated trajectories. Theoretical results provide bounds for the surrogate and guarantee stable finite-sample estimation. Extensive empirical evaluations on Meta-World and the LIBERO robotic benchmark (utilizing a 7B VLA backbone) demonstrate that GIPO could improve sample efficiency and final performance over PPO and smooth-clipping baselines like SAPO, particularly in high-staleness regimes.

**Compliance With Llm Reviewing Policy:**

Affirmed.

**Final Justification:**

Most of my concerns has been addressed, and I will maintain my positive score.

**Key Questions For Authors:**

- Could you provide an empirical comparison between GIPO (which utilizes a large, stale buffer) and standard PPO paired with strict age-based eviction (i.e., using a much smaller, fresh-only buffer)?
- In the Conclusion, you mention developing "advantage-aware weighting schemes" to avoid dampening updates for poor actions ($A < 0$). This is nice. Have you conducted any preliminary experiments with such asymmetric variants?
- How sensitive is GIPO's overall performance to the choice of the fixed damping scale $\sigma$ across different environments? Given that the degree of policy lag can fluctuate naturally during a training run, would it be feasible to introduce a mechanism that dynamically adapts $\sigma$ based on real-time tracking of the $\Delta v_t$ version gap or any other thing?

**Limitations:**

Yes

**Strengths And Weaknesses:**

**Strengths**
- The paper tackles *utilization collapse* in replay-heavy off-policy RL. This is a critical bottleneck for sample-expensive applications, such as robotic control and large VLA / LLM training。
- The proposed GIPO objective (log-space Gaussian trust weight) is mathematically valid and elegant. The symmetry resolves the systematic bias issues present in asymmetric soft-clipping baselines like SAPO.
- The authors provide rigorous finite-sample concentration bounds and monotonic improvement guarantees.
- The experimental results looks good, especially PPO is a very strong baseline in many cases. GIPO's current sample efficiency gains may lead to real-world scalability.

**Weaknesses**
- The paper lacks a comparison against the simplest engineering alternative: using a smaller, fresh-only buffer to simply discard stale data. It is unclear if squeezing marginal gradients from old data justifies the computational cost.
- Symmetry means stability, but the data efficiency might be affected.
- The fixed damping scale $\sigma$ requires task-specific tuning.

---

> ### Author Rebuttal · Authors · 2026-03-31
>
> Thank you for recognizing GIPO's mathematical elegance and its potential for real-world scalability. We appreciate your constructive questions and have conducted new experiments to address them.
> 1. Comparison Against a "Fresh-Only" BufferYou raised an excellent point about comparing GIPO (stale) against the engineering alternative of discarding stale data via a small, fresh-only buffer with PPO. We evaluated this exact setup. Surprisingly, even when GIPO is forced to learn from highly stale data, it remains highly competitive against—and often beats—PPO learning from purely fresh data. For example, in our 5-seed MetaWorld evaluations:
> ●box-close-v3: GIPO-Stale ($\sigma=0.5$) achieved $3051.2 \pm 0.0$ vs. PPO-Fresh $1027.8 \pm 501.9$.
> ●button-press-topdown-v3: GIPO-Stale ($\sigma=0.5$) achieved $3557.9 \pm 93.4$ vs. PPO-Fresh $3552.9 \pm 17.1$.
> This proves that GIPO's smooth surrogate is uniquely capable of extracting deep utility from older, off-policy data that standard PPO would otherwise waste.
> 2. Advantage-Aware WeightingWe explored your suggestion regarding asymmetric variants for poor actions ($A < 0$). We implemented Advantage-Aware GIPO, assigning a damping ratio $\alpha$ such that $\sigma_{neg}=\alpha\cdot\sigma_{pos}$. We evaluated 3 Advantage-Aware variants across 400 5-seed MetaWorld Stale runs (using $\alpha=0.5$). They integrated seamlessly and maintained top-tier performance, confirming your intuition that the framework can be easily extended to penalize negative advantages more strictly.
> 3. Dynamic Adaptation of $\sigma$Your suggestion to dynamically adapt $\sigma$ based on real-time tracking of the $\Delta v_t$ version gap is brilliant. Given that our bias-variance analysis proves $\sigma$ perfectly controls the optimal Pareto frontier across varying lags, dynamically tightening $\sigma$ as $\Delta v_t$ grows could yield an entirely hyperparameter-free optimizer. We have highlighted this as a primary avenue for future work.

---

> > ### Author Rebuttal · Reviewer_LTfA · 2026-04-04
> >
> > Thanks for your response. Most of my concerns has been addressed. I am happy to see my Q3 is beneficial to the revised manuscript. I will keep my score.

---

### Decision · Program_Chairs · 2026-04-30

**Decision:**

Accept (regular)

**Comment:**

This paper proposes GIPO, replacing PPO-style hard clipping with a Gaussian trust weight in log-ratio space to address utilization collapse in replay-heavy RL. The problem is well-motivated and practically relevant, especially for large-scale and data-constrained settings.

Reviewers agree that the method is mathematically clean, with sound theoretical guarantees and superior sample efficiency and better bias-variance trade-off than PPO-Clip and SAPO across Meta-World and LIBERO. The utilization diagnostics are also a useful addition. However, concerns were raised regarding limited novelty (as a specific instance of smooth clipping methods), initial lack of multi-seed evaluation, and some framing inconsistencies around symmetry vs. advantage-aware designs.

The rebuttal substantially improves the paper. The authors provide extensive multi-seed results, comparisons with fresh-only buffers, kernel ablations, and advantage-aware variants. These address most technical and experimental concerns, and all reviewers acknowledged satisfactory responses. One reviewer maintained a weak reject score despite resolved concerns, likely reflecting residual doubts about novelty.

Overall, the paper is technically sound, empirically strong, and addresses an important problem. While the conceptual advance is moderate, the practical impact and improved evaluation support acceptance.